# Multi-turn Reinforcement Learning
# from Preference Human Feedback

**Lior Shani** * 1
liorshani@google.com

**Aviv Rosenberg** * 1
avivros@google.com

**Asaf Cassel** * 1 3
acassel@mail.tau.ac.il

**Oran Lang** 1    **Daniele Calandriello** 2    **Avital Zipori** 1    **Hila Noga** 1    **Orgad Keller** 1

**Bilal Piot** 2    **Idan Szpektor** 1    **Avinatan Hassidim** 1    **Yossi Matias** 1    **Rémi Munos** 2

## Abstract

Reinforcement Learning from Human Feedback (RLHF) has become the standard approach for aligning Large Language Models (LLMs) with human preferences, allowing LLMs to demonstrate remarkable abilities in various tasks. Existing methods work by emulating the preferences at the single decision (turn) level, limiting their capabilities in settings that require planning or multi-turn interactions to achieve a long-term goal. In this paper, we address this issue by developing novel methods for Reinforcement Learning (RL) from preference feedback between two full multi-turn conversations. In the tabular setting, we present a novel mirror-descent-based policy optimization algorithm for the general multi-turn preference-based RL problem, and prove its convergence to Nash equilibrium. To evaluate performance, we create a new environment, Education Dialogue, where a teacher agent guides a student in learning a random topic, and show that a deep RL variant of our algorithm outperforms RLHF baselines. Finally, we show that in an environment with explicit rewards, our algorithm recovers the same performance as a reward-based RL baseline, despite relying solely on a weaker preference signal.

## 1  Introduction

A pinnacle of human intelligence is the ability to communicate with an environment, forming complex interactions to accomplish challenging goals. Dialogues are one example of such dynamic communication, where one party reacts to signals from the other parties and dynamically plans ahead to steer communication towards their purpose. Recent years have seen scientific breakthroughs in developing Large Language Models (LLMs) that can communicate with humans in natural language [Ouyang et al., 2022, Anil et al., 2023, Touvron et al., 2023, OpenAI, 2024, Google, 2024]. In order to align these models to human needs, many efforts have been made to train them with a given human feedback. In some concordance with human learning, this is usually achieved by reinforcing behaviors that align with the feedback, using a technique now called Reinforcement Learning from Human Feedback (RLHF; Christiano et al. [2017], Ziegler et al. [2019], Stiennon et al. [2020]).

RLHF methods build on the long studied field of Reinforcement Learning (RL), which focuses on learning optimal actions through reward feedback (a numerical signal) from the environment. However, defining a suitable reward function is challenging, leading to the common practice of collecting human preferences between choices. In the absence of rewards, a mapping from preference to reward is typically assumed in the form of the Bradely-Terry (BT; Bradley and Terry [1952]) model [Stiennon et al., 2020, Rafailov et al., 2023], enabling the use of a wide-variety of well-researched

---

* Equal contribution. 1 Google Research. 2 Google DeepMind. 3 Tel Aviv University.

38th Conference on Neural Information Processing Systems (NeurIPS 2024).

RL techniques. Alternatively, recent research [Munos et al., 2023, Azar et al., 2023, Tang et al., 2024] suggests a more direct use of preferences for learning, eliminating the need for this potentially limiting assumption.

Still, so far the main focus of both the RLHF and the direct preference learning literature was on single-turn scenarios, where given relevant context, the LLM generates one response and receives an immediate feedback that reflects its alignment quality. Importantly, while single-turn RLHF already provides significant gains for valuable AI systems, it lacks the adaptive and long-term capabilities that make human communication such a powerful tool, and usually characterize RL methods. This is especially apparent in temporally extended tasks, such as multi-turn dialogue [Irvine et al., 2023], complex tool use [Wang et al., 2022] and multi-step games [Hendrycks et al., 2022].

**Contributions.** In this work, we focus on improving the communication of AI agents with dynamic environments. To this end, we first extend the RLHF paradigm to the multi-turn setting, where the agent has a series of exchanges with an external (stochastic) environment (Section 3). Importantly, we consider (human) feedback that compares entire multi-turn conversations as opposed to single-turn scenarios, which compare individual actions on a per-turn basis. Conversation-level feedback allows to capture the long-term effect of individual actions, which may not be immediately apparent, and thus hard to define through turn-level feedback. For example, a seller agent asking too high a price may seem immediately bad, but becomes potentially good as part of a complete strategy to increase sale price. This difference is apparent in our preference model, making it better suited for multi-turn interactions.

Formalizing the multi-turn setting as a Contextual Markov Decision Process with end of interaction preference feedback, we devise several theoretically grounded algorithms (Section 4). Our main algorithm, Multi-turn Preference Optimization (MTPO), is a new policy optimization algorithm for the general multi-turn preference-based setting. MTPO is based on the Mirror Descent (MD) method [Nemirovskij and Yudin, 1983, Beck and Teboulle, 2003] together with self-play [Silver et al., 2017], and is proven to converge to a Nash equilibrium [Nash et al., 1950], i.e., a policy which is preferred over any other policy. We prove similar results for MTPO$-\tau$, a slight variant of MTPO that, similarly to Munos et al. [2023], uses a geometric mixture policy which interpolates the agent's policy with a fixed reference policy (with mixing rate $\tau$). These algorithms utilize a new form of preference-based Q-function that accounts for the long-term consequences of individual actions. Finally, leveraging our theoretical framework, we modify this Q-function to create a multi-turn RLHF algorithm and prove its convergence to an optimal policy (w.r.t the learned reward function).

We complement our theoretical findings with a policy-gradient version of our multi-turn algorithms for deep learning architectures (Section 4.1). To validate our approach, we apply our algorithms to train a T5 encoder-decoder LLM [Raffel et al., 2020], aiming to enhance its multi-turn dialogue abilities (Sections 5 and 6). We test our approach in a scenario without explicit rewards, where conversation quality is evaluated solely through preferences. To that end, we create a new environment called Education Dialogue, where a teacher guides a student in learning a random topic, by prompting Gemini [Team et al., 2023]. The conversation is judged based on preference feedback, using a constitution that defines effective learning [Bai et al., 2022, Lee et al., 2023] (see Section 5). In this environment, our multi-turn algorithms significantly outperform single-turn baselines, and our direct multi-turn preference approach outperforms multi-turn RLHF (Section 6). As an additional contribution, we publicly release the data of Education Dialogue.[1] Finally, we demonstrate that even in a reward-based environment, our preference-based algorithm achieves comparable performance to learning directly from rewards, as in standard RL, despite using a weaker signal. For this experiment, we utilize the LMRL-Gym [Abdulhai et al., 2023] Car Dealer environment, simulating a conversation where the agent (car dealer) aims to maximize the sale price.

**Related work.** Most related to our work is the RLHF literature, which aims to improve an LLM policy using preference data collected from humans. Earlier methods model a proxy reward [Ouyang et al., 2022] or preference function [Zhao et al., 2022], and apply traditional RL techniques. More recent methods directly optimize the policy [Rafailov et al., 2023, Azar et al., 2023, Tang et al., 2024, Song et al., 2024, Ethayarajh et al., 2024]. Another line of work, which forms the basis for MTPO, extends RLHF to games, aiming to compute a Nash equilibrium instead of an optimal policy w.r.t a fixed reward/preference. This includes Nash-MD [Munos et al., 2023], self-play and mixtures of the

---

[1] `https://github.com/google-research-datasets/Education-Dialogue-Dataset`

two like IPO-MD [Calandriello et al., 2024]. Nonetheless, these methods only consider single-turn problems, whereas MTPO provides the first guarantees for multi-turn settings. Note the difference from concurrent attempts to extend direct preference optimization to the token level [Rafailov et al., 2024], while a true multi-turn approach must deal with the additional uncontrollable tokens generated by the human in-between agent turns.

More broadly, preference-based RL (see survey by Wirth et al. [2017]) studies feedback in terms of preferences over two alternatives rather than absolute rewards. Feedback can be provided in various ways, e.g., at the level of states (turn-level), or entire trajectories [Chen et al., 2022, Saha et al., 2023, Wu and Sun, 2023, Wang et al., 2023, Zhan et al., 2023a,b]. The focus of this work is last state feedback which is an instance of trajectory feedback. Another closely related model is RL with aggregate feedback where only the sum of rewards in a trajectory is revealed to the agent [Efroni et al., 2021, Cohen et al., 2021, Chatterji et al., 2021, Cassel et al., 2024].

Lastly, there is vast literature on using RL to improve natural language generation for dialogue systems. The pioneering work of Li et al. [2016] focuses on designing rewards to capture important dialogue attributes such as semantic coherence and ease of answering. Other works tackle task-oriented dialogue, using RL to enhance the agent's ability to solve the underlying dialogue task [Wei et al., 2018]. Instead, this works takes a more general and fundamental approach, focusing on the algorithmic process of aligning an agent which repeatedly interacts with an environment. While dialogue systems are a promising application of our approach, as suggested by the experimental results of this paper (Sections 5 and 6), our algorithmic approach is much broader, including processes such as tool-use, reasoning, and many other applications that require aligning a complex multi-turn agent with human preferences.

## 2  Preliminaries

The interaction between an AI agent and its environment is captured in the fundamental contextual RL model, where the context is the initial prompt, the states are conversation summaries, and the actions are responses.

**Contextual Markov decision process.** A finite-horizon contextual Markov decision process (CMDP) $\mathcal{M}$ is defined by a tuple $(\mathcal{C}, \mathcal{X}, \mathcal{Y}, H, x_1, \rho_c, p)$ where $\mathcal{C}$ is the context space, $\mathcal{X}$ is the state space, $\mathcal{Y}$ is the action space, $H$ is the horizon, $x_1 \in \mathcal{X}$ is the initial state, $\rho_c \in \Delta_{\mathcal{C}}$ is the context distribution and $p : \mathcal{C} \times \mathcal{X} \times \mathcal{Y} \to \Delta_{\mathcal{X}}$ is the transition function such that $p(x' \mid c, x, y)$ is the probability to transition to state $x'$ after taking action $y$ in state $x$, given context $c$.

An interaction between the agent and the CMDP environment proceeds in $H$ steps. First, a context $c \in \mathcal{C}$ is sampled from $\rho_c$, and then the agent begins in the initial state $x_1$. In step $h \in [H]$, the agent observes the current state $x_h \in \mathcal{X}$, picks an action $y_h \in \mathcal{Y}$ and transitions to the next state $x_{h+1}$ sampled from the transition function $p(\cdot \mid c, x_h, y_h)$. At the end of the interaction, the agent arrives in a final state $x_{H+1}$. For simplicity, we assume that the state space can be decomposed into $H + 1$ disjoint subsets $\mathcal{X} = \uplus_{h=1}^{H+1} X_h$ such that, in step $h$ of the interaction, the agent is in some state $x_h \in \mathcal{X}_h$. A policy $\pi : \mathcal{C} \times \mathcal{X} \to \Delta_{\mathcal{Y}}$ is a mapping from a context and state to a distribution over actions. Together with transition $p$, $\pi$ induces a distribution over trajectories denoted by $\Pr_{\pi,p}[\cdot]$ (and $\mathbb{E}_{\pi,p}[\cdot]$ for the expectation), in which the trajectory is generated by sampling the actions according to the policy and next states according to the environment.

### 2.1  Single-turn Reinforcement Learning from Human Feedback

Unlike standard RL where the agent observes reward feedback for its actions, the influential work of Christiano et al. [2017] suggests to leverage preference data. In the single-turn setting, the agent generates a single sequence $y \in \mathcal{Y}$ given a context $c \in \mathcal{C}$. This is modeled as a Contextual Multi-Armed Bandit (CMAB), which is a CMDP instance with horizon $H = 1$. The feedback is given in the form of preference between two generated sequences. Formally, there exists a preference model $\mathcal{P} : \mathcal{C} \times \mathcal{Y} \times \mathcal{Y} \to [0, 1]$ such that $\mathcal{P}(y \succ y' \mid c)$ gives the probability that $y$ is preferred over $y'$ given context $c$. Preferences naturally extend to policies via expectation $\mathcal{P}(\pi \succ \pi' \mid c) = \mathbb{E}_{y \sim \pi(\cdot|c), y' \sim \pi'(\cdot|c)}[\mathcal{P}(y \succ y' \mid c)]$.

**Reinforcement Learning from Human Feedback.** In RLHF, it is assumed that there is a hidden reward function $r : \mathcal{C} \times \mathcal{Y} \to \mathbb{R}$ that defines the preferences through the Bradely-Terry (BT) model, i.e., $\mathcal{P}(y > y' \mid c) = \sigma(r(c, y) - r(c, y'))$, where $\sigma$ is the sigmoid. To reconstruct the reward, the RL algorithm is used to optimize the ELO score of the chosen action $y$ using a cross-entropy loss. This technique was adapted to RL fine-tuning LLMs [Ziegler et al., 2019], and has become the standard approach for aligning LLMs to human feedback [Stiennon et al., 2020, Rafailov et al., 2023].

**Learning from direct preferences.** Recently Munos et al. [2023], Azar et al. [2023] suggested to drop the BT assumption, and learn a direct preference model instead of reward. Munos et al. [2023] propose the Nash-MD algorithm which converges to the Nash equilibrium of a (regularized) preference model, i.e., a policy which is preferred over any other policy. In iteration $t + 1$, Nash-MD updates its policy $\pi_{t+1}$ using a mirror descent (MD) step projected to a geometric mixture policy. The mixture policy $\pi_t^\alpha(\cdot \mid c) \propto \pi_t(\cdot \mid c)^{1-\alpha\eta_t} \mu(\cdot \mid c)^{\alpha\eta_t}$ interpolates between the policy $\pi_t$ and a reference policy $\mu$, given a regularization coefficient $\alpha > 0$. Formally, for learning rate $\eta_t > 0$,

$$\pi_{t+1}(\cdot \mid c) = \arg \max_{\pi(\cdot \mid c) \in \Delta_{\mathcal{Y}}} \eta_t \mathcal{P}(\pi > \pi_t^\alpha \mid c) - \mathrm{KL}(\pi \| \pi_t^\alpha)[c] \qquad \forall c \in \mathcal{C},$$

where $\mathrm{KL}(\pi \| \pi')[c] \triangleq \mathrm{KL}(\pi(\cdot \mid c) \| \pi'(\cdot \mid c)) = \sum_y \pi(y \mid c) \log \frac{\pi(y \mid c)}{\pi'(y \mid c)}$.

# 3 Multi-turn Preference-Based RL

In the multi-turn setting, the agent repeatedly interacts with an external environment, an interaction we formulate using the CMDP model. Similarly to the single-turn case, we consider preference-based RL, where the feedback is given as preference instead of reward. However, in our case, we assume preferences are between final CMDP states with a shared initial context. Formally, there exists a preference model $\mathcal{P} : \mathcal{C} \times \mathcal{X}_{H+1} \times \mathcal{X}_{H+1} \to [0, 1]$ such that $\mathcal{P}(x_{H+1} > x'_{H+1} \mid c)$ gives the probability that $x_{H+1}$ is preferred over $x'_{H+1}$ given context $c$. That is, in order to receive feedback, a learning algorithm performs two interactions with the environment and observes a Bernoulli sample for which one is preferred. We follow the natural assumption of Munos et al. [2023] that the preference model is symmetric, i.e., $\mathcal{P}(x'_{H+1} > x_{H+1} \mid c) = 1 - \mathcal{P}(x_{H+1} > x'_{H+1} \mid c)$. We define the preference between a final state and a policy by $\mathcal{P}(x > \pi \mid c) = \mathbb{E}_{\pi,p}[\mathcal{P}(x > x_{H+1} \mid c)]$. Similarly, $\mathcal{P}(\pi > \pi' \mid c) = \mathbb{E}_{\pi,p}[\mathcal{P}(x_{H+1} > \pi' \mid c)]$. For brevity, since contexts are independent, we omit the context throughout the rest of the paper. Similarly to Munos et al. [2023], our objective is to find a policy $\pi^\star$ which is preferred over any other alternative policy, i.e.,

$$\pi^\star \in \arg \max_\pi \min_{\pi'} \mathcal{P}(\pi > \pi'),$$

which is a Nash equilibrium in the above two-player game defined by the preference model, following the minimax theorem [Von Neumann, 1928] (see Lemma 3.2). Notably, due to the anti-symmetric nature of the preference objective, the Nash equilibrium will have both agents following the same policy, and thus can be expressed as a single policy.

**Regularized preference model.** In the rest of the paper, we will consider a regularized version of the preference model. This is motivated by practical RLHF algorithms [Stiennon et al., 2020], and generalizes the single-turn model in Munos et al. [2023]. Let $\mu$ be a reference policy, and define the $\alpha$-regularized preference model as follows:

$$\mathcal{P}_\alpha(\pi > \pi') = \mathcal{P}(\pi > \pi') - \alpha \mathrm{KL}_p(\pi \| \mu) + \alpha \mathrm{KL}_p(\pi' \| \mu),$$

where $\mathrm{KL}_p(\cdot \| \cdot)$ is the KL-divergence between the distributions that the policies induce over trajectories in the CMDP. We prove the following two results for the regularized preference model. First, its KL term has a value difference-like decomposition into the KL-divergences at individual states. Second, it has a unique Nash equilibrium (proofs in Appendix A).

**Lemma 3.1.** *Let $\pi, \pi'$ be two policies, then:* $\mathrm{KL}_p(\pi \| \pi') = \mathbb{E}_{\pi,p}\big[\sum_{h=1}^H \mathrm{KL}(\pi \| \pi')[x_h]\big]$.

**Lemma 3.2.** *There exists a unique Nash equilibrium of the regularized preference model $\mathcal{P}_\alpha$.*

**Trajectory-wise vs. turn-wise preference feedback.** A naive adaptation of single-turn RLHF to the multi-turn scenario would treat each turn as a separate single-turn problem. This would require feedback for the preference between two actions in each turn. Instead, in the setting we consider, the

preference feedback is only between two full trajectories. Note that a single feedback for the entire trajectory is much more natural when considering conversations, since only the full conversations tell whether the objective was reached. Moreover, collecting preference data for intermediate actions could lead to destructive biases because the quality of an action can change dramatically depending on the actions taken later in the trajectory. For example, a chatbot directly answering a user query is usually a required behavior. Yet, when the chatbot does not have sufficient information to respond well, asking the user for more details might be a better action. Consequently, it is very hard for a rater to know which of these actions is better without observing how the conversation unrolls, i.e., without observing the user's reaction to the chatbot's question, and how the chatbot's response changes given this reaction. This difference demonstrates the challenge of multi-turn RL as it requires planning ahead instead of myopic reward maximization, which is the approach for single-turn RL.

**The multi-turn setting in LLMs.** While we consider a general preference-based RL setup (through the CMDP model), our focus is on applying this framework to multi-turn language-based interactions. The action space $\mathcal{Y}$ is a sequence of tokens of a vocabulary $\mathcal{V}$, and the state space at step $h$, $\mathcal{X}_h$, is a sequence based on the past sequences. For example, in conversational dialogues, the state $x_h$ holds the whole dialogue up to the $h$-th turn, the action $y_h$ is the current sequence generated by the agent, and the next-state is simply the concatenation of the conversation $x_h$ with the new $y_h$ and a next sequence sampled by the environment (the user's response). Alternatively, in the complex tool-use case, where an agent repeatedly interacts with different APIs, the current state includes the original user query and a summary of results from APIs so far, the action is a new API call or user-facing response, and next state is a new sequence summarizing previous state with the new API response.

**Remark 3.3** (Token-level application to the single-turn auto-regressive case)**.** *Notably, this formulation also captures the single-turn auto-regressive case. Clearly, this holds when considering only one turn, $H = 1$, but it ignores the token-level optimization done at each turn. Instead, we frame the auto-regressive problem by limiting the actions at each step to single vocabulary tokens, $\mathcal{Y} = \mathcal{V}$, and assuming a null deterministic environment ($x_{h+1}$ is the concatenation of $x_h$ and $y_h$). Importantly, our results apply to the token-level, which is usually neglected when devising single-turn algorithms.*

## 4   Algorithms for the multi-turn setting

**Preference-based Q-function.** Our algorithms rely on a fundamental concept in RL – the value and Q-functions. In reward-based RL, it is essential to define the value $V^\pi : \mathcal{X} \to \mathbb{R}$ as the expected reward when playing policy $\pi$ starting in some state $x_h$, i.e., $V^\pi(x_h) = \mathbb{E}_{\pi,p}\big[\sum_{h'=h}^H r(x_{h'}, y_{h'}) \mid x_h\big]$. In the preference-based scenario, we argue that value functions remain a powerful tool, even though there is no reward to maximize. We define the following regularized preference-based value functions, which are key to our algorithm.

$$Q_\alpha^{\pi,\pi'}(x_h, y_h) = \mathbb{E}_{\pi,p}\Big[\mathcal{P}(x_{H+1} \succ \pi') - \alpha \sum_{h'=h}^H \mathrm{KL}(\pi\|\mu)[x_{h'}] \mid x_h, y_h\Big],$$

$$V_\alpha^{\pi,\pi'}(x_h) = \mathbb{E}_{\pi,p}\Big[\mathcal{P}(x_{H+1} \succ \pi') - \alpha \sum_{h'=h}^H \mathrm{KL}(\pi\|\mu)[x_{h'}] \mid x_h\Big].$$

There are a few interesting points in the definition above. First, note that these values are functions of two policies $\pi, \pi'$. This is because the quality of a policy $\pi$ cannot be measured on its own, and must be compared to another policy $\pi'$. Second, while $\pi$ starts at the state $x_h$, the comparison policy $\pi'$ starts its trajectory from the initial state. This is a significant difference from the usual paradigm of Q-functions in RL and might seem peculiar at first glance. However, this formulation captures the fact that the optimal policy in a state should be preferred not only over any other policy along the sub-tree starting from this state, but also over any other policy, even ones that do not pass through this state at all. Although different in concept, the following lemma shows that our preference-based Q-function satisfies a value difference lemma, allowing us to optimize the policy locally in order to maximize our global objective (proof in Appendix B).

**Lemma 4.1.** *Let $\pi, \pi', \bar{\pi}$ be policies, then the following value difference lemma holds:*

$$\mathcal{P}_\alpha(\pi \succ \bar{\pi}) - \mathcal{P}_\alpha(\pi' \succ \bar{\pi}) = \mathbb{E}_{\pi',p}\Big[\sum_{h=1}^H \langle \pi - \pi', Q_\alpha^{\pi,\bar{\pi}}\rangle[x_h] + \alpha\mathrm{KL}(\pi'\|\mu)[x_h] - \alpha\mathrm{KL}(\pi\|\mu)[x_h]\Big],$$

*where $\langle \pi - \pi', Q\rangle[x] \triangleq \langle \pi(\cdot \mid x) - \pi'(\cdot \mid x), Q(x, \cdot)\rangle$ and $\langle x, y\rangle = \sum_i x(i)y(i)$ is the inner product.*

**MTPO.** We present the MTPO (Multi-turn Preference Optimization) algorithm, which provably solves the multi-turn preference-based RL objective. Formally, we prove MTPO converges to the unique Nash equilibrium of the regularized preference model. MTPO is based on two key principles: First, the regularized preference model defines a two-player anti-symmetric constant-sum game which can be solved using a self-play mirror descent method [Munos et al., 2023, Calandriello et al., 2024]. Second, our introduced Q-function allows to reduce the (global) optimization of the game into local mirror descent optimization problems in each state. Together they yield the MTPO update rule for iteration $(t+1)$,

$$\pi_{t+1}(\cdot \mid x_h) = \arg\max_\pi \eta_t \langle \pi, Q_\alpha^{\pi_t, \pi_t} \rangle[x_h] - \alpha\eta_t \text{KL}(\pi \| \mu)[x_h] - (1 - \alpha\eta_t)\text{KL}(\pi \| \pi_t)[x_h], \quad (1)$$

where $\eta_t$ is a learning rate. The solution can be made explicit in the following form (Appendix E.2):

$$\pi_{t+1}(y_h \mid x_h) \propto \mu(y_h \mid x_h)^{\alpha\eta_t} \pi_t(y_h \mid x_h)^{1-\alpha\eta_t} e^{\eta_t Q_\alpha^{\pi_t, \pi_t}(x_h, y_h)}. \quad (2)$$

The intuition behind the algorithm is observed nicely in this update rule – we improve the current policy in the direction of the regularized preference against itself (represented by the self-play Q-function), while not deviating too much and keeping close to the reference policy. The following is our main theoretical result: last-iterate convergence to Nash equilibrium (proof in Appendix B).

**Theorem 4.2.** *Let $\pi_\alpha^\star$ be the Nash equilibrium of the regularized preference model, and $\mathbb{Q}$ be a bound on the magnitude of the Q-functions. Then, for $\eta_t = \frac{2}{\alpha(t+2)}$, MTPO guarantees at every iteration t,*

$$\text{KL}_p(\pi_\alpha^\star \| \pi_t) \leq \frac{32H\mathbb{Q}^2}{\alpha^2(t+1)}.$$

*Let $\mu_{\min}$ be the minimal non-zero probability assigned by $\mu$, then $\mathbb{Q} \leq \max\{4\alpha H \log \frac{1}{\mu_{\min}}, 1\}$.*

*Proof sketch.* By Lemma 3.1, the global $\text{KL}_p(\pi_\alpha^\star \| \pi_{t+1})$ can be decomposed to the local KL in each state $x_h$, $\text{KL}(\pi_\alpha^\star \| \pi_{t+1})[x_h]$. Then, we use MD analysis in each state to bound the local KL as:

$$\text{KL}(\pi_\alpha^\star \| \pi_{t+1})[x_h] \leq (1 - \eta_t\alpha)\text{KL}(\pi_\alpha^\star \| \pi_t)[x_h] + 2\eta_t^2\mathbb{Q}^2$$
$$+ \eta_t(\langle \pi_t - \pi_\alpha^\star, Q_\alpha^{\pi_t, \pi_t} \rangle[x_h] + \alpha\text{KL}(\pi_\alpha^\star \| \mu)[x_h] - \alpha\text{KL}(\pi_t \| \mu)[x_h]),$$

giving a recursive guarantee dependent on the local one-step regularized advantage of the current policy against the Nash policy and an additional term bounded by $\mathbb{Q}$. We plug this local bound into the KL decomposition (Lemma 3.1), which gathers the local KL terms back to $\text{KL}_p(\pi_\alpha^\star \| \pi_t)$. Importantly, the value difference lemma (Lemma 4.1) aggregates the advantage terms to the global regularized preference $\mathcal{P}_\alpha(\pi_t > \pi_t) - \mathcal{P}_\alpha(\pi_\alpha^\star > \pi_t)$, which is non-positive by the optimality of $\pi_\alpha^\star$. This leaves us with the global recursive bound: $\text{KL}_p(\pi_\alpha^\star \| \pi_{t+1}) \leq (1 - \eta_t\alpha)\text{KL}_p(\pi_\alpha^\star \| \pi_t) + 2\eta_t^2\mathbb{Q}^2$. We conclude by unrolling the recursion with the chosen $\eta_t$. $\square$

**MTPO with mixture policy.** Inspired by Nash-MD [Munos et al., 2023], we present a variant of MTPO which makes use of the mixture policy $\pi_t^\alpha(\cdot \mid x) \propto \pi_t(\cdot \mid x)^{1-\alpha\eta_t} \mu(\cdot \mid x)^{\alpha\eta_t}$. This variant, which we call MTPO-$\tau$ (where $\tau$ will be the mixing coefficient in our experiments), gives similar theoretical guarantees (see Theorem B.2 in Appendix B) and performs better in practice (see Section 6). In fact, the following MTPO-$\tau$ update rule is almost equivalent to MTPO (Equation (1)) with the only difference being the policies that define the Q-function, $Q_\alpha^{\pi_t, \pi_t}$ vs. $Q_\alpha^{\pi_t^\alpha, \pi_t^\alpha}$.

$$\pi_{t+1}(\cdot \mid x_h) = \arg\max_\pi \eta_t \langle \pi, Q_\alpha^{\pi_t^\alpha, \pi_t^\alpha} \rangle[x_h] - \text{KL}(\pi \| \pi_t^\alpha)[x_h].$$

MTPO-$\tau$ naturally extends Nash-MD to the multi-turn setting, and reveals that Nash-MD is a self-play algorithm itself, but plays $\pi_t^\alpha$ instead of $\pi_t$. Practically, MTPO has the computational advantage over MTPO-$\tau$ (and Nash-MD) of not keeping the additional policy $\pi_t^\alpha$. Moreover, MTPO avoids the difficulty of computing the geometric mixture, which Munos et al. [2023] approximate heuristically via linear interpolation between the logits of the two policies.

**Multi-turn RLHF.** While we focused so far on our preference-based algorithms, our derivation holds for any online reward function since it is built on the mirror-descent method. Specifically, in the case of multi-turn RLHF, we consider the reward function $r^{\mathrm{RLHF}}$ learned from preference data using the Bradley-Terry model, and define the corresponding regularized Q-function $Q_\alpha^{\pi,\mathrm{RLHF}}(x_h, y_h) = \mathbb{E}_{\pi,p}\big[r^{\mathrm{RLHF}}(x_{H+1}) - \alpha \sum_{h'=h}^{H} \mathrm{KL}(\pi\|\mu)[x_{h'}] \mid x_h, y_h\big]$. By replacing $Q_\alpha^{\pi_t,\pi_t}$ in Equation (1) with $Q_\alpha^{\pi_t,\mathrm{RLHF}}$, we obtain the multi-turn RLHF algorithm that converges to the regular RLHF objective – the optimal regularized policy w.r.t. the reward $r^{\mathrm{RLHF}}$ (see Theorem B.6 in Appendix B). This complementary contribution emphasizes the similarity and difference between the RLHF and MTPO algorithms: The optimization process is identical for both methods, with the exception that the RLHF reward is fixed and computed w.r.t. the data policy, whereas preference-based MTPO uses an adaptive self-play mechanism to compute preferences w.r.t. the current policy.

### 4.1 Deep RL implementation

Our deep RL implementation is a natural adaptation of the tabular algorithms presented in the previous section. At each iteration, training data is acquired by sampling a batch of contexts from the data, and using each context to sample two trajectories with the current policy $\pi_{\theta_t}$. Then, the final states of both trajectories serve as inputs to a direct preference model that outputs the probability of one being preferred over the other. Similarly to the way the preference model is trained in Nash-MD [Munos et al., 2023], this preference model is trained in advance on the available offline preference data.

The update rule in Equation (1) relies on the $Q$-function of the current policy. We therefore use an actor-critic policy optimization based approach and train two models, a policy $\pi_\theta$, and its value $V_{\alpha,\phi}^{\pi_{\theta_t},\pi_{\theta_t}}$, which is typically used to estimate the advantage, $A_\alpha^{\pi,\pi'}(x,y) \triangleq Q_\alpha^{\pi,\pi'}(x,y) - V_\alpha^{\pi,\pi'}(x)$ [Schulman et al., 2017]. For simplicity and computational efficiency, we implement a policy-gradient (PG) based approach and ignore the MD stability term $\mathrm{KL}(\pi_{\theta_t}\|\pi_{t-1})$, similarly to the implementation of the Nash-MD algorithm. We justify this simplification with the fact that the KL regularization w.r.t. the fixed reference policy $\mu$ already provides stability to our online algorithm, somewhat similarly to the way the Follow-The-Regularized-Leader (FTRL; Orabona [2019]) algorithm operates. Nevertheless, we believe that this additional MD penalty should contribute to the performance and stability of the algorithm, as shown in [Tomar et al., 2020], and we leave this for further research. This yields the following losses, when action $y$ is played at state $x$,

$$\mathcal{L}_{\mathrm{policy}}(\theta;x,y) = -\hat{A}_\alpha^{\pi_{\theta_t},\pi_{\theta_t}}(x,y)\log\pi_\theta(y\mid x) + \alpha\mathrm{KL}(\pi_\theta\|\mu)[x],$$

$$\mathcal{L}_{\mathrm{value}}(\phi;x) \quad = \left(\hat{V}_\alpha^{\pi_{\theta_t},\pi_{\theta_t}}(x) - V_{\alpha,\phi}^{\pi_{\theta_t},\pi_{\theta_t}}(x)\right)^2,$$

where $\hat{V}_\alpha^{\pi_{\theta_t},\pi_{\theta_t}}, \hat{A}_\alpha^{\pi_{\theta_t},\pi_{\theta_t}}$, are estimations of the current value and advantage using Generalized Advantage Estimation (GAE, Schulman et al. [2017]). We also batch-normalize the value-loss and advantage as recommended in [Andrychowicz et al., 2020]. Finally, when the policy is an auto-regressive language model, which generates actions token-by-token until an end-of-sequence signal is generated, we use a turn-level value (and not a token-level value as done in [Stiennon et al., 2020]). That is, the value model gets as input a state represented by a sequence of tokens, and outputs a single scalar value instead of a scalar value for each token in the action sequence. This is justified by our analysis which treats whole turns as single actions. We leave the many ways to combine turn-level and token-level values for future research.

## 5 Experimental Setup

This section describes the domains and models used in our experiments. To create online environments suited for multi-turn RL, we mimic the RLHF process [Stiennon et al., 2020], replacing the human parts with prompted state-of-the-art LLMs, similarly to Abdulhai et al. [2023] (see Figure 1):

1. **Dataset creation:** First, we devise a story-line for the user and the environment, describing their characters and goals. Then, we generate a dataset by prompting a state-of-the-art LLM such as Gemini [Team et al., 2023] or GPT [Brown et al., 2020] with the story-line. When generating data, a full conversation is sampled at once, meaning that both the agent and environment are generated together to make them more consistent. Furthermore, to create

**Conversation Generation Process**

**Prompt:** Simulate a conversation between a teacher in school and a student. There is a small chance that the teacher is successful in teaching the student so he understands the topic. The conversation lasts roughly 10-15 turns but ends when either side says [end of conversation]. The teacher wants to teach the student about {Topic}. The student likes {Student preference}. The teacher does not know that beforehand. The student prefers to learn this way, {Student reactions}. The teacher likes {Teacher Preference}. He prefers to teach this way, {Teacher reaction}.
Output the conversation and the probability that the student understood the material, in the following format.
Conversation: [ Teacher: "...", Student: "...", Teacher: "...", Student: "...",] Probability: "...",

**Topic:** The Periodic Table
**Student preference:** interactive learning/class discussions/asking questions    **Student reactions:** and gets anxious otherwise
**Teacher preference:** direct instruction/lecture-based learning    **Teacher reactions:** and might insist on teaching this way

**Interaction 1**

**Teacher:** Today, we're going to learn about the periodic table.
**Student:** I'm not sure I understand. Can we discuss it as a class?
**Teacher:** No, I prefer to lecture. The periodic table is a tabular arrangement of the chemical elements.
**Student:** I'm feeling a little anxious. I learn better through interactive discussions.
**Teacher:** Class discussions are a waste of time. The periodic table is organized into rows and columns.
⋮
**Student:** I'm overwhelmed. I don't think I can learn this way.
**Teacher:** It's your own fault if you don't understand. You're not paying attention.
**Student:** I am paying attention! I just need a different teaching style.
**Teacher:** Well, I can't change my entire teaching style for one student.
**Student:** Maybe we could try a different approach?
**Teacher:** No. I've been teaching for years. I know what I'm doing.
**Student:** Well, I don't like it. [end of conversation]

**Interaction 2**

**Teacher:** Today, we're going to learn about the Periodic Table.
**Student:** Cool! How are we going to learn about it?
**Teacher:** I'm going to explain it to you and then we'll have a quiz.
**Student:** That sounds a bit boring...
**Teacher:** It's important to understand the basics first.
**Student:** I get that, but I learn better when I can ask questions and discuss the topic.
**Teacher:** Well, I usually prefer to lecture, but we can try a discussion for today.
⋮
**Student:** That's interesting! So, how do you know which element has a specific number of proton?
**Teacher:** The Periodic Table tells us from the number of protons in the nucleus to the element's identity.
**Student:** I think I get it now. Thanks for explaining it in a way that I could understand!
**Teacher:** You're welcome. Let's move on to the quiz now.
**Student:** [end of conversation]

**Preference Reward**

**Prompt:** You are an expert at assessing teachers. Here are two interactions between a teacher and a student.
Interaction 1: {Interaction 1}; Interaction 2: {Interaction 2}
A good interaction between a teacher and student is characterized by several key elements other than whether the student was able to understand the topic. The teacher should present information clearly and enthusiastically, encouraging questions and active participation. Students should feel comfortable asking for clarification, offering their own insights, and respectfully challenging ideas.
Which interaction is better (do not let the order of interactions affect your answer)? Output 1 or 2.
**Output:** 2

Figure 1: Education Dialogue data generation process. *Top:* prompt used to generate conversation with Gemini. *Middle:* conversations sampled from the the interaction between the teacher and student models. *Bottom*: prompt used for the preference oracle.

> a diverse set of conversations, we devise a diverse list of attributes for both the agent and environment, sample attributes out of the list, and pass it to the generation prompt.

2. **Environment preparation:** Once the data is curated, we use it to fine-tune two smaller LLMs, one for the agent and one for the environment, using teacher forcing.

3. **Preference/reward learning:** We prepare preference data by sampling pairs of conversations from the agent and environment models. To label the data, we prompt a high-capacity LLM with either instructions on how to score a conversation, or criteria for preferring a conversation over another. The data is used to fine-tune two smaller LLMs: an RLHF reward model (with BT loss), and a preference model (with probability regression loss).

We experiment with two domains, preference-based *Education Dialogue* and reward-based *Car Dealer*:

**Education Dialogue.** The core of our approach is learning when there is no clear reward, instead only (human) preferences can be acquired. To validate our approach in this scenario, we created a novel multi-turn task for evaluating algorithms based on preference data. In this scenario, which we term Education Dialogue, a teacher (agent) is faced with the task of teaching a student (environment) a given topic in the best means possible. We follow the dataset creation procedure and prompt Gemini Ultra [Google, 2024] to create such interactions between the teacher and student. The teacher is prompted with a learning topic in science, history, etc. The student is prompted with the characteristics of its learning habits, e.g., prefers interactive learning, lecture-based learning or hands-on activities. The preference model is prompted with instructions that define a good learning interaction. For reproducibility, and to further advance the research of the multi-turn setting, we openly release the data and prompts used to create this new benchmark.[1] For more details, see Appendix C and the example in Figure 1.

Table 1: Side-by-side evaluation for Education Dialogue using Flan-T5 XL as the prompted preference model. Each entry is the average preference of 1,600 conversations generated with row method $y$, over ones generated with column method $y'$. We evaluate each method using 3 different seeds, compute 3 × 3 comparisons matrix and report the mean (the standard deviation is reported in Appendix D).

| | SL | Single-turn-reward | | Single-turn-value | | Multi-turn | | |
| --- | --- | --- | --- | --- | --- | --- | --- | --- |
| | SFT | RLHF | Nash | RLHF | Nash | RLHF | MTPO | MTPO-$\tau$ |
| SFT | – | 0.164 | 0.347 | 0.197 | 0.324 | 0.212 | 0.091 | 0.093 |
| RLHF-reward | 0.836 | – | 0.628 | 0.515 | 0.654 | 0.399 | 0.392 | 0.354 |
| Nash-reward | 0.653 | 0.372 | – | 0.411 | 0.51 | 0.328 | 0.281 | 0.242 |
| RLHF-value | 0.803 | 0.485 | 0.589 | – | 0.568 | 0.408 | 0.396 | 0.366 |
| Nash-value | 0.676 | 0.346 | 0.49 | 0.432 | – | 0.45 | 0.298 | 0.27 |
| RLHF-multi | 0.788 | 0.601 | 0.672 | 0.592 | 0.55 | – | 0.433 | 0.412 |
| MTPO | 0.909 | 0.608 | 0.719 | 0.604 | 0.702 | 0.567 | – | 0.439 |
| MTPO-$\tau$ | **0.907** | **0.646** | **0.758** | **0.634** | **0.73** | **0.588** | **0.561** | – |

**Car Dealer.** In this LMRL-Gym [Abdulhai et al., 2023] domain, a car dealer is assigned with the task of selling a car at the highest price to a customer. We skip the data creation step, and directly use the Car Dealer published data to fine-tune the dealer (agent) and customer (environment) T5-large models. The reward is calculated by prompting a Flan-T5 model to extract the sale price from the conversation, whenever a sale has occurred. When using a preference-based algorithm, the preference of one trajectory over the other is computed using the BT model with the rewards of the two trajectories.

**Single-turn baselines.** The key hypothesis of this work is that conversation-level signals are preferred over single-turn signals for optimizing multi-turn trajectories. To verify this in the Education Dialogue domain, we devise two single-turn baselines by sampling data where each conversation turn has two different policy responses. The first baseline, called *single-turn-reward*, rates the two responses using a modified preference prompt (see Appendix C), in which the model is asked to evaluate the responses by their effect on the overall conversation. This technique is prevalent when human raters are asked to evaluate multi-turn data. The second baseline, called *single-turn-value*, assumes access to a Monte-Carlo estimate of the value: it uses our original preference prompt (see Figure 1) by continuing the trajectories of both possibilities and then calculating the preference in the end. For both baselines, we train an RLHF algorithm and a preference-based Nash-MD algorithm.

**Models.** The agent and environment are modeled with T5 encoder-decoder models. Specifically, we use the T5-large (770M) and T5-XL (3B) models. The same models are used for the RLHF BT-based reward and preference-based models. For prompted reward/preference models, we make use of the Flan-T5 XL (3B) [Chung et al., 2024]. For training, we use a configuration of 4 × 4 Tensor Processing Units (TPUs; Jouppi et al. [2023]) which typically yields 0.1 training steps per second, where a step consists of learning a 10-turn episode. A detailed list of hyperparameters is found in Appendix D. We run each evaluation on 1600 random samples from an independent evaluation set.

## 6 Experiments

In this section we evaluate the algorithms proposed in Section 4. We start with the preference-based Education Dialogue environment (see Section 5), and compare our multi-turn algorithms to SFT (supervised fine-tuning) as well as single-turn baselines. We note that, unlike single-turn benchmarks which are based on data with real human preferences, our golden preference data itself is generated by an LLM (Gemini Ultra). Therefore, the true goal in our curated environment is to align the model with the preference of this highly capable LLM rather than a human rater. While human evaluation is always interesting, here it is actually only a proxy to alignment with the data distribution. To efficiently validate our models, we start with a thorough comparison between our baselines and candidates using a prompted Flan-T5 XL model as a judge, which was verified to correlate with the high-capacity Gemini Ultra (Table 1). We then compare our best candidates using the same Gemini Ultra which generated the preference alignment feedback (Table 2).

Table 2: Side-by-side evaluation for Education Dialogue using Gemini Ultra as the prompted preference model. Each entry is the average preference of 1,000 conversations generated with row method $y$, over ones generated with column method $y'$.

| | | SL | Single-turn | Multi-turn | |
| --- | --- | --- | --- | --- | --- |
| | | SFT | RLHF-reward | RLHF | MTPO-$\tau$ |
| T5-Large (770M) | SFT | – | 0.206 | 0.164 | 0.086 |
| | RLHF-reward | 0.794 | – | 0.452 | 0.277 |
| | RLHF-multi | 0.836 | 0.548 | – | 0.288 |
| | MTPO-$\tau$ | **0.914** | **0.723** | **0.712** | – |
| T5-XL (3B) | SFT | – | 0.295 | 0.101 | 0.041 |
| | RLHF-reward | 0.705 | – | 0.180 | 0.069 |
| | RLHF-multi | 0.899 | 0.82 | – | 0.139 |
| | MTPO-$\tau$ | **0.959** | **0.951** | **0.861** | – |

Table 3: Car Dealer experiments averaged across 5 seeds and reported with 95% confidence interval.

| | Online oracle | | Model from preferences data | |
| --- | --- | --- | --- | --- |
| | Reward (RL) | MTPO | RLHF | MTPO |
| Reward (Price) | 58.4 (0.3) | 57.1 (0.2) | 53.2 (0.3) | 58.6 (0.3) |

**Multi vs. single turn.** Tables 1 and 2 show that all multi-turn algorithms (MTPO and multi-turn RLHF) with conversation-level feedback significantly outperform the single-turn baselines, validating our hypothesis. We conjecture that it is attributed to several factors: First, the effect of a single-decision on the whole conversation is hard to capture, causing highly inaccurate single-turn reward/preference models. Notably, this leads to inferior performance of Nash-MD compared to single-turn RLHF, since it optimizes to find Nash equilibrium of this inaccurate model while RLHF does not stray so far from the reference. Second, even if one could estimate the current policy's value, this estimate becomes biased when the policy changes during training. Finally, single-turn preferences consider only "local" decisions which share the same conversational path, and not how these decisions "globally" compare to other possible paths, as captured by the preference-based Q-function $Q_\alpha^{\pi_t,\pi_t}$ (see Section 4).

**MTPO vs. multi-turn RLHF.** Comparing our three multi-turn algorithms, we see two main results. First, the two variants of MTPO outperform multi-turn RLHF. This is expected since the environment is not reward-based, and hence it extends the results of [Munos et al., 2023, Calandriello et al., 2024] from the single-turn case, and supports the theoretical claim that MTPO converges to the Nash policy while multi-turn RLHF converges to the optimal policy w.r.t the learned reward (which is based only on the reference policy). Second, MTPO-$\tau$ outperforms MTPO. While both algorithms converge to the same Nash equilibrium, we conjecture that the superior performance of MTPO-$\tau$ stems from the stochasticity that the mixture policy $\pi_t^\alpha$ introduces. Namely, $\pi_t$ might tend towards deterministic behavior, causing less informative feedback from self-play, as the two sampled trajectories would be very similar. On the other hand, $\pi_t^\alpha$ is more stochastic, providing diversity in the sampled trajectories.

**Reward-based environment.** In an additional experiment, we test MTPO and multi-turn RLHF in the reward-based Car Dealer environment, where the goal is maximizing sale price (see Section 5). We compare a standard policy-gradient RL algorithm against our algorithms in two scenarios: an online scenario where the reward or preference feedback is given using an online oracle, and an RLHF-like setting, where we first create preference data using the oracle, and then use it to fine-tune a (BT) reward and preference models. Table 3 shows that even though MTPO receives preferences instead of the explicit optimization target (rewards), it still learns as good as RL. Interestingly, MTPO recovers a slightly higher reward than multi-turn RLHF despite the fact the true preferences are sampled from a BT model. This may imply that a preference model generalizes better than a BT-reward model, perhaps because it is independent of the sampling policy.

**Limitations.** This work presents a proof of concept for the potential of MTPO to improve existing single-turn techniques. Our experimental setup might be limited by the relatively small T5-based models and the use of prompt-based environments. We leave applications to state-of-the-art models and algorithms, and more realistic environments to future work.

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

# Appendix

## Table of Contents

# A Proofs for Section 3

## A.1 KL decomposition (proof of Lemma 3.1)

**Lemma** (restatement of Lemma 3.1). *Let $\pi, \pi'$ be two policies, then:*

$$\mathrm{KL}_p(\pi\|\pi') = \mathbb{E}_{\pi,p}\left[\sum_{h=1}^{H} \mathrm{KL}(\pi\|\pi')[x_h]\right].$$

*Proof.* $\mathrm{KL}_p(\cdot\|\cdot)$ is defined as the KL-divergence between the distributions that the policies induce over trajectories in the MDP (denoted by $\tau = (x_1, y_1, \ldots, x_H, y_H, x_{H+1})$, formally:

$$
\begin{aligned}
\mathrm{KL}_p(\pi\|\pi') &= \sum_{\tau} \Pr_{\pi,p}[\tau] \log \frac{\Pr_{\pi,p}[\tau]}{\Pr_{\pi',p}[\tau]} \\
&= \sum_{\tau} \Pr_{\pi,p}[\tau] \log \frac{\prod_{h=1}^{H} \pi(y_h \mid x_h) p(x_{h+1} \mid x_h, y_h)}{\prod_{h=1}^{H} \pi'(y_h \mid x_h) p(x_{h+1} \mid x_h, y_h)} \\
&= \sum_{\tau} \Pr_{\pi,p}[\tau] \log \prod_{h=1}^{H} \frac{\pi(y_h \mid x_h)}{\pi'(y_h \mid x_h)} \\
&= \mathbb{E}_{\pi,p}\left[\log \prod_{h=1}^{H} \frac{\pi(y_h \mid x_h)}{\pi'(y_h \mid x_h)}\right] \\
&= \mathbb{E}_{\pi,p}\left[\sum_{h=1}^{H} \log \frac{\pi(y_h \mid x_h)}{\pi'(y_h \mid x_h)}\right] \\
&= \mathbb{E}_{\pi,p}\left[\sum_{h=1}^{H} \mathrm{KL}(\pi(\cdot \mid x_h)\|\pi'(\cdot \mid x_h))\right]. \qquad \square
\end{aligned}
$$

## A.2 Existence and uniqueness of the Nash equilibrium (proof of Lemma 3.2)

**Lemma** (restatement of Lemma 3.2). *There exists a unique Nash equilibrium of the regularized preference model $\mathcal{P}_\alpha$.*

*Proof.* The existence of the Nash equilibrium is proved in Theorem A.1. In order to prove the uniqueness, we use the fact that, from Theorem 4.2, the algorithm MTPO produces a sequence of policies $\pi_t$ that converges to any Nash equilibrium $\pi^*$, in the sense that $\lim_{t\to\infty} \mathrm{KL}_p(\pi^*\|\pi_t) = 0$. From the definition of the KL-divergence between policies, we have

$$\mathrm{KL}_p(\pi^*\|\pi_t) = \mathbb{E}_{\pi^*,p}\left[\sum_{h=1}^{H} \mathrm{KL}(\pi^*\|\pi_t)[x_h]\right] = \sum_{x\in\mathcal{X}} \rho^{\pi^*}(x)\mathrm{KL}(\pi^*\|\pi_t)[x],$$

where $\rho^{\pi^*}(x)$ is the probability to reach $x$ when following $\pi^*$.

Now, the fixed point of the MTPO dynamics (Equation (2)) shows that any Nash equilibrium satisfies, for any $x \in X, y \in \mathcal{Y}$,

$$\pi^*(y \mid x) \propto \mu(y \mid x) e^{\frac{1}{\alpha} Q_\alpha^{\pi^*, \pi^*}(x,y)}.$$

In particular, we notice that any Nash equilibrium has the same support as $\mu$, thus the set of reachable states under $\mu$ is exactly the same set as the set of reachable states under any Nash equilibrium $\pi^*$.

So, from any reachable state $x \in \mathcal{X}$ (i.e., such that $\rho^\mu(x) > 0$), we have that the sequence $\pi_t(\cdot \mid x)$ converges (in KL-divergence) to the Nash equilibrium $\pi^*(\cdot \mid x)$. Since a single sequence cannot converge to two different values, we have that the Nash equilibrium $\pi^*(\cdot \mid x)$ is uniquely defined in that state. Since the behavior generated by a policy depends on the policy at the set of states that are reacheable only, and since we have seen that all Nash equilibria have the same set of reachable states, we deduce the uniqueness of the policy defined by a Nash equilibrium. $\qquad \square$

**Theorem A.1.** *The game defined by the payoff function $(\pi, \pi') \mapsto \mathcal{P}_\alpha(\pi \succ \pi')$ has a Nash equilibrium.*

*Proof.* First we prove that there exists (at least) one max-min policy $\pi^* \in \arg\max_\pi \min_{\pi'} \mathcal{P}_\alpha(\pi \succ \pi')$.

For any $\pi$, the map $\pi' \in \mathcal{X} \mapsto \mathcal{P}_\alpha(\pi \succ \pi')$ is continuous, thus upper semi-continuous (u.s.c.). We know that the pointwise minimum of u.s.c. functions is also u.s.c. (see, e.g. Aliprantis and Border [2006], Lemma 2.41). Thus the function $\pi \in \Pi \mapsto \min_{\pi'} \mathcal{P}_\alpha(\pi \succ \pi')$ is also u.s.c. and since $\Pi$ is compact, we can apply Theorem 2.43 of Aliprantis and Border [2006] to deduce that this function attains a maximum value in $\Pi$ and that the set of maximizers is compact. Thus there exists (at least) one policy denoted by $\pi^* \in \arg\max_{\pi \in \Pi} \min_{\pi' \in \Pi} \mathcal{P}_\alpha(\pi \succ \pi')$.

Now from Lemma A.2 we know that the regularized preference model $\mathcal{P}_\alpha(\pi \succ \pi')$ defines a concave-convexlike game. Also for any $\pi'$, the map $\pi \mapsto \mathcal{P}_\alpha(\pi \succ \pi')$ is u.s.c. Thus we can apply the minimax Theorem 4.2 of Sion [1958] to deduce that

$$\max_\pi \min_{\pi'} \mathcal{P}_\alpha(\pi \succ \pi') = \min_{\pi'} \max_\pi \mathcal{P}_\alpha(\pi \succ \pi').$$

We deduce from

$$\frac{1}{2} = \max_\pi \mathcal{P}_\alpha(\pi \succ \pi) \geq \max_\pi \min_{\pi'} \mathcal{P}_\alpha(\pi \succ \pi') = \min_{\pi'} \max_\pi \mathcal{P}_\alpha(\pi \succ \pi') \geq \min_{\pi'} \mathcal{P}_\alpha(\pi' \succ \pi') = \frac{1}{2},$$

that the value of the game is $1/2$, and that $\min_{\pi'} \mathcal{P}_\alpha(\pi^* \succ \pi') = 1/2$. Thus $(\pi^*, \pi^*)$ is a Nash equilibrium of the game defined by the regularized preference model $(\pi, \pi') \mapsto \mathcal{P}_\alpha(\pi \succ \pi')$. $\qquad\square$

**Lemma A.2.** *The mapping $(\pi, \pi') \mapsto \mathcal{P}_\alpha(\pi \succ \pi')$ is concave-convexlike, which, in the context of a symmetric preference model, means that for any couple of policies $(\pi_1, \pi_2)$ and any coefficient $c \in [0, 1]$, there exists a policy $\pi_c$ such that for any policy $\pi'$, we have*

$$c\mathcal{P}_\alpha(\pi_1 \succ \pi') + (1 - c)\mathcal{P}_\alpha(\pi_2 \succ \pi') \leq \mathcal{P}_\alpha(\pi_c \succ \pi').$$

*Proof.* Let us define the notion of reach probability: for any state $x_h \in \mathcal{X}_h$, let us write $\rho^\pi(x_h)$ the probability to reach the specific state $x_h \in \mathcal{X}_h$ when following $\pi$: $\rho^\pi(x_h) = \Pr_{\pi,p}[x_h]$. First notice we can represent the regularized preference $\mathcal{P}_\alpha(\pi \succ \pi')$ using reach probabilities:

$$\mathcal{P}_\alpha(\pi \succ \pi') = \sum_{x_{H+1}, x'_{H+1} \in \mathcal{X}_{H+1}} \rho^\pi(x_{H+1}) \rho^{\pi'}(x'_{H+1}) \mathcal{P}(x_{H+1} \succ x'_{H+1})$$

$$- \alpha \sum_{h=1}^{H} \sum_{x_h \in \mathcal{X}_h} \rho^\pi(x_h) \mathrm{KL}(\pi \| \mu)[x_h] - \rho^{\pi'}(x_h) \mathrm{KL}(\pi' \| \mu)[x_h].$$

Now, consider two policies $\pi_1$ and $\pi_2$ and a coefficient $c \in [0, 1]$. From Lemma A.3 we have that there exists a policy $\pi_c$ such that for any $x_h$, we have $\rho^{\pi_c}(x_h) = c\rho^{\pi_1}(x_h) + (1 - c)\rho^{\pi_2}(x_h)$. We can write

$$\mathcal{P}_\alpha(\pi_c \succ \pi') = \sum_{x_{H+1}, x'_{H+1} \in \mathcal{X}_{H+1}} \left[ c\rho^{\pi_1}(x_{H+1}) + (1 - c)\rho^{\pi_2}(x_{H+1}) \right] \rho^{\pi'}(x'_{H+1}) \mathcal{P}(x_{H+1} \succ x'_{H+1})$$

$$- \alpha \sum_{h=1}^{H} \sum_{x_h \in \mathcal{X}_h} \left[ c\rho^{\pi_1}(x_h) + (1 - c)\rho^{\pi_2}(x_h) \right] \mathrm{KL}(\pi_c \| \mu)[x_h]$$

$$+ \alpha \sum_{h=1}^{H} \sum_{x_h \in \mathcal{X}_h} \rho^{\pi'}(x_h) \mathrm{KL}(\pi' \| \mu)[x_h]$$

$$= c\mathcal{P}(\pi_1 \succ \pi') + (1 - c)\mathcal{P}(\pi_2 \succ \pi')$$

$$- \alpha \sum_{h=1}^{H} \sum_{x_h \in \mathcal{X}_h} \left[ c\rho^{\pi_1}(x_h) + (1 - c)\rho^{\pi_2}(x_h) \right] \mathrm{KL}(\pi_c \| \mu)[x_h]$$

$$+ \alpha \sum_{h=1}^{H} \sum_{x_h \in \mathcal{X}_h} \rho^{\pi'}(x_h) \mathrm{KL}(\pi' \| \mu)[x_h].$$

Now from the convexity of $\pi \mapsto \mathrm{KL}(\pi \| \mu)[x_h]$, and the definition of $\pi_c$, we have that

$$\rho^{\pi_c}(x_h) \mathrm{KL}(\pi_c \| \mu)[x_h] = \left[ c\rho^{\pi_1}(x_h) + (1 - c)\rho^{\pi_2}(x_h) \right] \mathrm{KL}(\pi_c \| \mu)[x_h]$$

$$\leq c\rho^{\pi_1}(x_h) \mathrm{KL}(\pi_1 \| \mu)[x_h] + (1 - c)\rho^{\pi_2}(x_h) \mathrm{KL}(\pi_2 \| \mu)[x_h].$$

Thus

$$\mathcal{P}_\alpha(\pi_c > \pi') \geq c\mathcal{P}(\pi_1 > \pi') + (1-c)\mathcal{P}(\pi_2 > \pi')$$

$$- \alpha \sum_{h=1}^{H} \sum_{x_h \in \mathcal{X}_h} c\rho^{\pi_1}(x_h)\mathrm{KL}(\pi_1\|\mu)[x_h] + (1-c)\rho^{\pi_2}(x_h)\mathrm{KL}(\pi_2\|\mu)[x_h]$$

$$+ \alpha \sum_{h=1}^{H} \sum_{x_h \in \mathcal{X}_h} c\rho^{\pi'}(x_h)\mathrm{KL}(\pi'\|\mu)[x_h] + (1-c)\rho^{\pi'}(x_h)\mathrm{KL}(\pi'\|\mu)[x_h]$$

$$= c\mathcal{P}_\alpha(\pi_1 > \pi') + (1-c)\mathcal{P}_\alpha(\pi_2 > \pi'). \qquad \square$$

**Lemma A.3.** *For any state $x_h \in \mathcal{X}_h$, let us write $\rho^\pi(x_h)$ the probability to reach the specific state $x_h \in \mathcal{X}_h$ when following $\pi$, i.e., $\rho^\pi(x_h) = \mathrm{Pr}_{\pi,p}[x_h]$. Then, for any two policies $(\pi_1, \pi_2)$ and any coefficient $c \in [0,1]$, there exists a policy $\pi_c$ such that for any $h = 1, \ldots, H+1$, and any $x_h \in \mathcal{X}_h$, we have*

$$\rho^{\pi_c}(x_h) = c\rho^{\pi_1}(x_h) + (1-c)\rho^{\pi_2}(x_h).$$

*Proof.* Let us define $\pi_c$ as a function of $\pi_1$, $\pi_2$, and $c$: for any $x \in \mathcal{X}$, $y \in \mathcal{Y}$,

$$\pi_c(y \mid x) = \frac{c\rho^{\pi_1}(x)\pi_1(y \mid x) + (1-c)\rho^{\pi_2}(x)\pi_2(y \mid x)}{c\rho^{\pi_1}(x) + (1-c)\rho^{\pi_2}(x)}.$$

We now prove the lemma by induction on $h$. It holds for $h = 1$ since we have

$$\rho^{\pi_c}(x_1) = 1 = c + (1-c) = c\rho^{\pi_1}(x_1) + (1-c)\rho^{\pi_2}(x_1).$$

Now assume the claim holds for any $x_h \in \mathcal{X}_h$, then for any $x_{h+1} \in \mathcal{X}_{h+1}$,

$$\rho^{\pi_c}(x_{h+1}) = \sum_{x_h \in \mathcal{X}_h} \rho^{\pi_c}(x_h) \sum_{y_h \in \mathcal{Y}_h} p(x_{h+1} \mid x_h, y_h)\pi_c(y_h \mid x_h)$$

$$= \sum_{x_h \in \mathcal{X}_h} \rho^{\pi_c}(x_h) \sum_{y_h \in \mathcal{Y}_h} p(x_{h+1} \mid x_h, y_h)\frac{c\rho^{\pi_1}(x_h)\pi_1(y_h \mid x_h) + (1-c)\rho^{\pi_2}(x_h)\pi_2(y_h \mid x_h)}{c\rho^{\pi_1}(x_h) + (1-c)\rho^{\pi_2}(x_h)}$$

$$= \sum_{x_h \in \mathcal{X}_h} \rho^{\pi_c}(x_h) \sum_{y_h \in \mathcal{Y}_h} p(x_{h+1} \mid x_h, y_h)\frac{c\rho^{\pi_1}(x_h)\pi_1(y_h \mid x_h) + (1-c)\rho^{\pi_2}(x_h)\pi_2(y_h \mid x_h)}{\rho^{\pi_c}(x_h)}$$

$$= \sum_{x_h \in \mathcal{X}_h} \sum_{y_h \in \mathcal{Y}_h} p(x_{h+1} \mid x_h, y_h)\left[c\rho^{\pi_1}(x_h)\pi_1(y_h \mid x_h) + (1-c)\rho^{\pi_2}(x_h)\pi_2(y_h \mid x_h)\right]$$

$$= c\rho^{\pi_1}(x_{h+1}) + (1-c)\rho^{\pi_2}(x_{h+1}),$$

where the third inequality is by the induction hypothesis. $\qquad \square$

# B Proofs for section 4

## B.1 Regularized preference-based Q-function (proof of Lemma 4.1)

**Lemma** (restatement of Lemma 4.1)**.** *Let $\pi, \pi'$ be two policies. For every $x_{H+1} \in \mathcal{X}_{H+1}$, it holds that $V_\alpha^{\pi,\pi'}(x_{H+1}) = \mathcal{P}(x_{H+1} > \pi')$. Furthermore, for every $h \in [H]$ the following recursive relations hold:*

$$V_\alpha^{\pi,\pi'}(x_h) = \mathbb{E}_{y_h \sim \pi(\cdot|x_h)} Q_\alpha^{\pi,\pi'}(x_h, y_h),$$

$$Q_\alpha^{\pi,\pi'}(x_h, y_h) = \mathbb{E}_{x_{h+1} \sim p(\cdot|x_h,y_h)} V_\alpha^{\pi,\pi'}(x_{h+1}) - \alpha \mathrm{KL}(\pi \| \mu)[x_h].$$

*Moreover, let $\bar{\pi}$ be a third policy, then the following value difference lemma holds:*

$$\mathcal{P}_\alpha(\pi > \bar{\pi}) - \mathcal{P}_\alpha(\pi' > \bar{\pi}) = \mathbb{E}_{\pi',p}\Big[\sum_{h=1}^H \langle \pi - \pi', Q_\alpha^{\pi,\bar{\pi}} \rangle[x_h] + \alpha \mathrm{KL}(\pi' \| \mu)[x_h] - \alpha \mathrm{KL}(\pi \| \mu)[x_h]\Big].$$

*Proof.* We prove the lemma by casting the preference-based RL problem as an adversarial MDP, see Appendix E for the details. Set $r^t(x_{H+1}) = \mathcal{P}(x_{H+1} > \pi')$, then $Q_\alpha^{\pi,\pi'} = Q_\alpha^{\pi,t}$ (see Definition E.3 for the definition of the regularized Q-function). Now the lemma follows directly from Lemmas E.4 and E.5. □

## B.2 Convergence of MTPO (Proof of Theorem 4.2)

**Theorem** (restatement of Theorem 4.2)**.** *Let $\pi_\alpha^\star$ be the Nash equilibrium of the regularized preference model $\mathcal{P}_\alpha$. Then, for the choice $\eta_t = \frac{2}{\alpha(t+2)}$, MTPO guarantees at every iteration $t$ that*

$$\mathrm{KL}_p(\pi_\alpha^\star \| \pi_t) \le \frac{32 H \mathbb{Q}^2}{\alpha^2(t+1)},$$

*where $\mathbb{Q}$ is a bound on the magnitude of the Q-functions.*

*Proof.* The theorem follows directly from Theorem B.1. □

**Theorem B.1.** *Running the MTPO algorithm, we have that at every iteration $t$:*

$$\mathrm{KL}_p(\pi_\alpha^\star \| \pi_{t+1}) \le (1 - \eta_t \alpha) \mathrm{KL}_p(\pi_\alpha^\star \| \pi_t) + 2\eta_t^2 \mathbb{E}_{\pi_\alpha^\star, p}\left[\sum_{h=1}^H \left\| Q_\alpha^{\pi_t, \pi_t}(x_h, \cdot) - \alpha \log \frac{\pi_t(\cdot \mid x_h)}{\mu(\cdot \mid x_h)} \right\|_\infty^2\right].$$

*Thus, for the choice $\eta_t = \frac{2}{\alpha(t+2)}$ we have*

$$\mathrm{KL}_p(\pi_\alpha^\star \| \pi_t) \le \frac{8}{\alpha^2(t+1)} \cdot \max_t \mathbb{E}_{\pi_\alpha^\star, p}\left[\sum_{h=1}^H \left\| Q_\alpha^{\pi_t, \pi_t}(x_h, \cdot) - \alpha \log \frac{\pi_t(\cdot \mid x_h)}{\mu(\cdot \mid x_h)} \right\|_\infty^2\right].$$

*Finally,*

$$\mathrm{KL}_p(\pi_\alpha^\star \| \pi_t) \le \frac{32H}{t+1} \cdot \max\left\{\frac{1}{\alpha^2}, H^2 \log^2 \mu_{\min}\right\},$$

*where $\mu_{\min} = \min_{(x,y) \in \mathcal{X} \times \mathcal{Y} : \mu(y|x) > 0} \mu(y \mid x)$ is the minimal non-zero probability assigned by $\mu$.*

*Proof.* We prove the theorem by casting the preference-based RL problem as an adversarial MDP, see Appendix E for the details. Set $r^t(x_{H+1}) = \mathcal{P}(x_{H+1} > \pi_t)$, then $Q_\alpha^{\pi_t, \pi_t} = Q_\alpha^{\pi_t, t}$ (see Definition E.3 for the definition of the regularized Q-function). Now, MTPO is equivalent to running mirror descent policy optimization. Thus, by Lemma E.6 with $\pi = \pi_\alpha^\star$,

$$\mathrm{KL}_p(\pi_\alpha^\star \| \pi_{t+1}) \le (1 - \eta_t \alpha) \mathrm{KL}_p(\pi_\alpha^\star \| \pi_t) + 2\eta_t^2 \mathbb{E}_{\pi_\alpha^\star, p}\left[\sum_{h=1}^H \left\| Q_\alpha^{\pi_t, \pi_t}(x_h, \cdot) - \alpha \log \frac{\pi_t(\cdot \mid x_h)}{\mu(\cdot \mid x_h)} \right\|_\infty^2\right]$$

$$+ \eta_t(\mathcal{P}_\alpha(\pi_t > \pi_t) - \mathcal{P}_\alpha(\pi_\alpha^\star > \pi_t))$$

$$\le (1 - \eta_t \alpha) \mathrm{KL}_p(\pi_\alpha^\star \| \pi_t) + 2\eta_t^2 \mathbb{E}_{\pi_\alpha^\star, p}\left[\sum_{h=1}^H \left\| Q_\alpha^{\pi_t, \pi_t}(x_h, \cdot) - \alpha \log \frac{\pi_t(\cdot \mid x_h)}{\mu(\cdot \mid x_h)} \right\|_\infty^2\right],$$

where the second inequality optimality of $\pi_\alpha^\star$. The last claim follows directly from Lemma E.8. Similarly to Nash-MD Munos et al. [2023], this is the Nash-equilibrium of regularized preference model, $\mathcal{P}_\alpha$, following the minimax theorem [Von Neumann, 1928] □

## B.3 MTPO with mixture policy (MTPO-$\tau$)

Inspired by Nash-MD, we present a variant of MTPO which makes use of the mixture policy $\pi_t^\alpha$, which we call MTPO-$\tau$. Define the regularized policy $\pi_t^\alpha$ as a geometric mixture between the current policy $\pi_t$ and the reference policy $\mu$:

$$\pi_t^\alpha(y \mid x) = \frac{\pi_t(y \mid x)^{1-\eta_t\alpha}\mu(y \mid x)^{\eta_t\alpha}}{\sum_{y'\in\mathcal{Y}}\pi_t(y' \mid x)^{1-\eta_t\alpha}\mu(y' \mid x)^{\eta_t\alpha}}.$$

The MTPO-$\tau$ update rule is then:

$$\pi_{t+1}(\cdot \mid x_h) = \arg\max_\pi \eta_t\left\langle\pi, Q_\alpha^{\pi_t^\alpha,\pi_t^\alpha}\right\rangle[x_h] - \mathrm{KL}(\pi\|\pi_t^\alpha)[x_h] \qquad \forall h \in [H], x_h \in \mathcal{X}_h.$$

where $\mathrm{KL}(\cdot\|\cdot)$ is the standard KL-divergence. It is well-known that his optimization problem has the following explicit closed-form:

$$\pi_{t+1}(y_h \mid x_h) = \frac{\pi_t^\alpha(y_h \mid x_h)e^{\eta_t Q_\alpha^{\pi_t^\alpha,\pi_t^\alpha}(x_h,y_h)}}{\sum_{y_h'\in\mathcal{Y}}\pi_t^\alpha(y_h' \mid x_h)e^{\eta_t Q_\alpha^{\pi_t^\alpha,\pi_t^\alpha}(x_h,y_h')}} \qquad \forall h \in [H], x_h \in \mathcal{X}_h, y_h \in \mathcal{Y}.$$

Next, we show that MTPO-$\tau$ converges to the Nash equilibrium of the $\alpha$-regularized preference model.

**Theorem B.2.** *Let $\pi_\alpha^\star$ be the Nash equilibrium of the regularized preference model $\mathcal{P}_\alpha$. Then, for the choice $\eta_t = \frac{2}{\alpha(t+2)}$, MTPO-$\tau$ guarantees at every iteration $t$ that*

$$\mathrm{KL}_p(\pi_\alpha^\star\|\pi_t^\alpha) \le \frac{9H\mathbb{Q}^2}{\alpha^2(t+1)},$$

*where $\mathbb{Q}$ is a bound on the magnitude of the Q-functions.*

*Proof.* The theorem follows directly from Theorem B.3 and corollary B.4. $\qquad\square$

**Theorem B.3.** *Running the MTPO-$\tau$ algorithm, we have that at every iteration $t$:*

$$\mathrm{KL}_p(\pi_\alpha^\star\|\pi_{t+1}) \le (1-\eta_t\alpha)\mathrm{KL}_p(\pi_\alpha^\star\|\pi_t) + 2\eta_t^2\mathbb{E}_{\pi_\alpha^\star,p}\left[\sum_{h=1}^H \left\|Q_\alpha^{\pi_t^\alpha,\pi_t^\alpha}(x_h,\cdot)\right\|_\infty^2\right].$$

*Thus, for the choice $\eta_t = \frac{2}{\alpha(t+2)}$ we have*

$$\mathrm{KL}_p(\pi_\alpha^\star\|\pi_t) \le \frac{8}{\alpha^2(t+1)}\cdot\max_t\mathbb{E}_{\pi_\alpha^\star,p}\left[\sum_{h=1}^H \left\|Q_\alpha^{\pi_t^\alpha,\pi_t^\alpha}(x_h,\cdot)\right\|_\infty^2\right].$$

*Finally,*

$$\mathrm{KL}_p(\pi_\alpha^\star\|\pi_t) \le \frac{8H}{t+1}\cdot\max\left\{\frac{1}{\alpha^2}, H^2\log^2\mu_{\min}\right\}.$$

*Proof.* We prove the theorem by casting the preference-based RL problem as an adversarial MDP, see Appendix E for the details. Set $r^t(x_{H+1}) = \mathcal{P}(x_{H+1} > \pi_t^\alpha)$, then $Q_\alpha^{\pi_t^\alpha,\pi_t^\alpha} = Q_\alpha^{\pi_t^\alpha,t}$ (see Definition E.3 for the definition of the regularized Q-function). Now, Nash-MD is equivalent to running mixture mirror descent policy optimization. Thus, by Lemma E.7 with $\pi = \pi_\alpha^\star$,

$$\begin{aligned}\mathrm{KL}_p(\pi_\alpha^\star\|\pi_{t+1}) &\le (1-\eta_t\alpha)\mathrm{KL}_p(\pi_\alpha^\star\|\pi_t) + 2\eta_t^2\mathbb{E}_{\pi_\alpha^\star,p}\left[\sum_{h=1}^H \left\|Q_\alpha^{\pi_t^\alpha,\pi_t^\alpha}(x_h,\cdot)\right\|_\infty^2\right] \\ &\quad + \eta_t(\mathcal{P}_\alpha(\pi_t^\alpha > \pi_t^\alpha) - \mathcal{P}_\alpha(\pi_\alpha^\star > \pi_t^\alpha)) \\ &\le (1-\eta_t\alpha)\mathrm{KL}_p(\pi_\alpha^\star\|\pi_t) + 2\eta_t^2\mathbb{E}_{\pi_\alpha^\star,p}\left[\sum_{h=1}^H \left\|Q_\alpha^{\pi_t^\alpha,\pi_t^\alpha}(x_h,\cdot)\right\|_\infty^2\right],\end{aligned}$$

where the second inequality optimality of $\pi_\alpha^\star$. The last claim follows directly from Lemma E.9. $\quad\square$

**Corollary B.4.** *Running the MTPO-$\tau$ algorithm, we have that at every iteration $t$:*

$$\mathrm{KL}_p(\pi_\alpha^\star \| \pi_t^\alpha) \le (1 - \eta_t \alpha) \mathrm{KL}_p(\pi_\alpha^\star \| \pi_t) + \frac{\eta_t}{2}.$$

*Thus, for the choice $\eta_t = \frac{2}{\alpha(t+2)}$ we have*

$$\mathrm{KL}_p(\pi_\alpha^\star \| \pi_t^\alpha) \le \frac{9H}{t+1} \cdot \max\left\{\frac{1}{\alpha^2}, H^2 \log^2 \mu_{\min}\right\}.$$

*Proof.* By Munos et al. [2023, Lemma 1], for every $x_h \in \mathcal{X}_h$,

$$\mathrm{KL}(\pi_\alpha^\star \| \pi_t^\alpha)[x_h] \le (1 - \eta_t \alpha) \mathrm{KL}(\pi_\alpha^\star \| \pi_t)[x_h] + \eta_t \alpha \mathrm{KL}(\pi_\alpha^\star \| \mu)[x_h].$$

We finish the proof by taking the expectation $\mathbb{E}_{\pi_\alpha^\star, p}[\cdot]$ and using Lemmas 3.1 and B.5. The second part is by Theorem B.3. $\square$

**Lemma B.5.** *It holds that $\mathrm{KL}_p(\pi_\alpha^\star \| \mu) \le \frac{1}{2\alpha}$.*

*Proof.* By the optimality of $\pi_\alpha^\star$ we have that $\mathcal{P}_\alpha(\pi_\alpha^\star \succ \mu) \ge \mathcal{P}_\alpha(\pi_\alpha^\star \succ \pi_\alpha^\star)$. Now, since $\mathcal{P}_\alpha(\pi_\alpha^\star \succ \pi_\alpha^\star) = 1/2$, we get

$$\alpha \mathrm{KL}_p(\pi_\alpha^\star \| \mu) \le \mathcal{P}(\pi_\alpha^\star \succ \mu) - \frac{1}{2} \le 1 - \frac{1}{2} = \frac{1}{2}. \qquad \square$$

## B.4  Convergence of multi-turn RLHF

**Theorem B.6.** *Let $\pi_\alpha^{\star,\mathrm{RLHF}}(\cdot \mid x) = \arg\max_\pi V_\alpha^{\pi,\mathrm{RLHF}}(x)$ for every $x \in \mathcal{X}$. Then, for the choice $\eta_t = \frac{2}{\alpha(t+2)}$, multi-turn RLHF guarantees at every iteration $t$, $\mathrm{KL}_p(\pi_\alpha^{\star,\mathrm{RLHF}} \| \pi_t) \le \frac{32H\mathbb{Q}^2}{\alpha^2(t+1)}$.*

*Proof.* The theorem follows directly from Theorem B.7. $\square$

**Theorem B.7.** *Running the multi-turn RLHF algorithm, we have that at every iteration $t$:*

$$\mathrm{KL}_p(\pi_\alpha^{\star,\mathrm{RLHF}} \| \pi_{t+1}) \le (1 - \eta_t \alpha) \mathrm{KL}_p(\pi_\alpha^\star \| \pi_t) + 2\eta_t^2 \mathbb{E}_{\pi_\alpha^{\star,\mathrm{RLHF}}, p}\left[\sum_{h=1}^H \left\| Q_\alpha^{\pi_t, \pi_t}(x_h, \cdot) - \alpha \log \frac{\pi_t(\cdot \mid x_h)}{\mu(\cdot \mid x_h)} \right\|_\infty^2\right].$$

*Thus, for the choice $\eta_t = \frac{2}{\alpha(t+2)}$ we have*

$$\mathrm{KL}_p(\pi_\alpha^{\star,\mathrm{RLHF}} \| \pi_t) \le \frac{8}{\alpha^2(t+1)} \cdot \max_t \mathbb{E}_{\pi_\alpha^{\star,\mathrm{RLHF}}, p}\left[\sum_{h=1}^H \left\| Q_\alpha^{\pi_t, \pi_t}(x_h, \cdot) - \alpha \log \frac{\pi_t(\cdot \mid x_h)}{\mu(\cdot \mid x_h)} \right\|_\infty^2\right].$$

*Finally,*

$$\mathrm{KL}_p(\pi_\alpha^{\star,\mathrm{RLHF}} \| \pi_t) \le \frac{32H}{t+1} \cdot \max\left\{\frac{1}{\alpha^2}, H^2 \log^2 \mu_{\min}\right\},$$

*where $\mu_{\min} = \min_{(x,y) \in \mathcal{X} \times \mathcal{Y} : \mu(y|x) > 0} \mu(y \mid x)$ is the minimal non-zero probability assigned by $\mu$.*

*Proof.* We prove the theorem by casting the regularized RLHF problem as an adversarial MDP, see Appendix E for the details. Set $r^t(x_{H+1}) = r^{\mathrm{RLHF}}(x_{H+1})$, then $Q_\alpha^{\pi_t, \mathrm{RLHF}} = Q_\alpha^{\pi_t, t}$ (see Definition E.3 for the definition of the regularized Q-function). Now, multi-turn RLHF is equivalent to running mirror descent policy optimization. Thus, by Lemma E.6 with $\pi = \pi_\alpha^{\star,\mathrm{RLHF}}$,

$$\mathrm{KL}_p(\pi_\alpha^{\star,\mathrm{RLHF}} \| \pi_{t+1}) \le (1 - \eta_t \alpha) \mathrm{KL}_p(\pi_\alpha^{\star,\mathrm{RLHF}} \| \pi_t) + 2\eta_t^2 \mathbb{E}_{\pi_\alpha^{\star,\mathrm{RLHF}}, p}\left[\sum_{h=1}^H \left\| Q_\alpha^{\pi_t, \pi_t}(x_h, \cdot) - \alpha \log \frac{\pi_t(\cdot \mid x_h)}{\mu(\cdot \mid x_h)} \right\|_\infty^2\right]$$

$$+ \eta_t \left( V_\alpha^{\pi_t, \mathrm{RLHF}}(x_1) - V_\alpha^{\pi_\alpha^{\star,\mathrm{RLHF}}, \mathrm{RLHF}}(x_1) \right)$$

$$\le (1 - \eta_t \alpha) \mathrm{KL}_p(\pi_\alpha^\star \| \pi_t) + 2\eta_t^2 \mathbb{E}_{\pi_\alpha^\star, p}\left[\sum_{h=1}^H \left\| Q_\alpha^{\pi_t, \pi_t}(x_h, \cdot) - \alpha \log \frac{\pi_t(\cdot \mid x_h)}{\mu(\cdot \mid x_h)} \right\|_\infty^2\right],$$

where the second inequality optimality of $\pi_\alpha^{\star,\mathrm{RLHF}}$. The last claim follows directly from Lemma E.8. $\square$

# C   The Education Dialogue environment

## C.1   Prompts for creating the environment

We use the following prompt to generate conversations using Gemini [Google, 2024]:

```
Simulate a conversation between a teacher in school and
a student.  There is a small chance that the teacher is
successful in teaching the student so he understands the
topic.  The conversation lasts roughly 10-15 turns but ends
when either side says [end of conversation].  The teacher
wants to teach the student about {topic}.  The student likes
{student_pref}.  The teacher does not know that beforehand.
The student prefers to learn this way, {student_reaction}.
The teacher likes {teacher_pref}.  He prefers to teach this
way, {teacher_reaction}.  Output the conversation and the
probability that the student understood the material, in the
following format.
#
Conversation:
[
 Teacher:  "...",
 Student:  "...",
 Teacher:  "...",
 Student:  "...",
]
Probability:  "...",
#
```

The topic is sampled from the following topics list:

```
Photosynthesis, Evolution, DNA, Newton's First Law of
Motion, Newton's Second Law of Motion, Newton's Third Law
of Motion, Archimedes' Principle, Conservation of Energy,
Pythagorean Theorem, Allegory, Metaphor, Personification,
Foreshadowing, Irony, Atoms, Elements, Molecules, The
Periodic Table, The French Revolution, The Industrial
Revolution, The Russian Revolution, World War 1, World War
2, The American Civil War, The September 11th Attacks, The
Declaration of Independence, The Pyramids, The Parthenon,
The Colosseum, The Hagia Sophia, The Taj Mahal, The Great
Wall of China, The Machu Picchu, Angkor Wat, The Palace
of Versailles, The White House, The Tower of London, Notre
Dame Cathedral, The Eiffel TowerConfucius, Julius Caesar,
Leonardo da Vinci, William Shakespeare, Napoleon Bonaparte,
Abraham Lincoln, Albert Einstein, Martin Luther King,
Nelson Mandela, Marie Curie, Genghis Khan, Christopher
Columbus, Joan of Arc, Winston Churchill, Vincent van Gogh,
Pablo Picasso, Salvador Dali, The Roman Empire, The Cold
War, Zeus, Poseidon, Ares, Hercules, Achilles, Minotaur,
Medusa, The Solar System, The Big Bang, Supply and Demand,
Communism, Capitalism, Democracy, Dictatorship, Sigmund
Freud, Cells, The Circulatory System, The Respiratory System,
The Respiratory System, The Nervous System, Neurons
```

The student's learning preferences (`student_pref`) are sampled from the following list:

```
interactive learning/class discussions/asking
questions, direct instruction/lecture-based learning,
hands-on activities/real-world applications, creative
expression/story telling/gamification
```

The student's reactions to not learning in their preferred ways (`student_reaction`) are sampled from the following list:

```
and gets rude otherwise, and gets disengaged otherwise,
and gets frustrated otherwise, and gets anxious otherwise,
but might adapt to other methods, and might tell it to the
teacher
```

The teacher's teaching preferences (`teacher_pref`) are sampled from the following list:

```
direct instruction/lecture-based learning, interactive
learning/class discussions/inquiry-based learning,
experiential learning/hands-on activities, formative
assessment
```

The teacher's reactions to different learning methods (`teacher_reaction`) are sampled from the following list:

```
and gets frustrated otherwise, and blames the student
otherwise, and gives up otherwise, but might adapt to the
student, and might insist on teaching this way
```

We use the following prompt to query Gemini for the preference between two conversations (`conv1` and `conv2`):

```
You are an expert at assesing teachers.  Here are two
interactions between a teacher and a student.
#
Interaction 1:
{conv1}
#
Interaction 2:
{conv2}
#
A good interaction between a teacher and student is
characterized by several key elements other than whether
the student was able to understand the topic.  The teacher
should present information clearly and enthusiastically,
encouraging questions and active participation.  Students
should feel comfortable asking for clarification, offering
their own insights, and respectfully challenging ideas.
Which interaction is better (do not let the order
interactions affect your answer)?  Output 1 or 2.
```

For the single-turn baseline, we use the following modified prompt to query Gemini for preferences:

```
You are an expert at assesing teachers.  Here is an
interaction between a teacher and a student.
#
Interaction:
{conv}
#
Here are two possible responses by the teacher:
#
Response 1:
{resp1}
#
Response 2:
{resp2}
#
A good interaction between a teacher and student is
characterized by several key elements other than whether
the student was able to understand the topic.  The teacher
should present information clearly and enthusiastically,
encouraging questions and active participation.  Students
should feel comfortable asking for clarification, offering
their own insights, and respectfully challenging ideas.
Assuming that the teacher and student continue the
interaction with one of these responses, which response
will lead to a better interaction (do not let the order
interactions affect your answer)?  Output 1 or 2.
```

## C.2  Examples of interactions

1. Teacher: Good morning. Today, we're going to talk about personification, a literary device that gives human qualities to nonhuman things.

   Student: That sounds a bit abstract. Can you give me an example?

   Teacher: Sure. 'The wind whispered through the trees.'

   Student: But the wind can't actually whisper.

   Teacher: That's correct. But we're using personification to create a more vivid image in our minds. Let's try an activity. I'll give you a sentence, and you try to personify it.

   Student: Okay, that sounds fun.

   Teacher: The car purred down the street.

   Student: The car roared down the street!

   Teacher: Excellent! You're getting the hang of it. Now, let's talk about why writers use personification.

   Student: I'm not really understanding why they do.

   Teacher: That's okay. Let's brainstorm. Can you think of any reasons why authors might use personification?

   Student: Maybe to make the story more interesting?

   Teacher: That's a great reason. Anything else?

   Student: Maybe to make it more relatable?

   Teacher: Yes, personification can help us relate to objects and ideas on a deeper level. Do you think you understand personification now?

   Student: I think so. It's a creative way to make things come alive.

   Teacher: Exactly. I'm glad you enjoyed learning about it. Let's do a quick review before we end the class.

   Student: Sounds good.

   Teacher: Great. I'll ask you a few questions, and you can answer as if you were a personifying writer.

   Student: Okay.

2. Teacher: Today, we're going to learn about The Solar System.

Student: Sounds boring. Can we do something more hands-on?

Teacher: I have a great activity planned. We're going to build a scale model of the Solar System using different-sized balls.

Student: That sounds cool!

Teacher: First, let's talk about the planets. There are eight planets in our Solar System.

Student: I know some of them, like Earth and Mars.

Teacher: Excellent! Now, let's start building our model. We'll use these balls to represent the planets and arrange them in order from the Sun.

Student: This is fun! I can see how big the planets are compared to each other.

Teacher: I'm glad you're enjoying it. Now, let's talk about the moons and other objects that orbit the planets.

Student: Can we build a model of the moons?

Teacher: Absolutely! This will help you understand their different characteristics.

Student: I think I get it now. This is a great way to learn!

Teacher: I'm so glad to hear that. Let's finish building our model and then we'll review what we've learned.

Student: Okay.

Teacher: Well done! I believe you have a good understanding of The Solar System now.

Student: Thanks! I learned a lot. You're welcome. [end of conversation]

3. Teacher: Today, we're going to learn about World War 1.

   Student: Okay.

   Teacher: The war began in 1914 when Archduke Franz Ferdinand of Austria-Hungary was assassinated.

   Student: I've heard of him.

   Teacher: The war was fought between two main alliances: the Allies and the Central Powers.

   Student: Can you tell me more about the Allies?

   Teacher: Sure. The Allies included France, Russia, and Great Britain.

   Student: What were the Central Powers?

   Teacher: They were Germany, Austria-Hungary, and Ottoman Russia.

   Student: This is a lot of information. Can we slow down a bit?

   Teacher: Of course. Let's review what we've learned so far.

   Student: I'm not sure I understand everything yet.

   Teacher: That's okay. We can go over it again. But first, I want you to try to summarize what we've learned so far.

   Student: World War 1 started when Archduke Ferdinand was assassinated, and the Allies and Central Powers fought each other.

   Teacher: Good. Now, let's go over the key events of the war.

   Student: Okay.

   Teacher: The war ended in 1918 with the defeat of the Central Powers.

   Student: I think I understand it now.

   Teacher: Great! I'm glad you do.

   Student: Thanks, I appreciate it. You're welcome. [end of conversation]

4. Teacher: Today, we're going to learn about Machu Picchu, the ancient Incan city.

   Student: Can you just tell me the facts? I don't need all this extra stuff.

   Teacher: Sure. Machu Picchu was built in the 15th century in the Andes Mountains of Peru.

   Student: What made it so special?

   Teacher: Its location on a mountain ridge provided stunning views of the surrounding landscape.

   Student: That's it?

   Teacher: Well, there's more to it. Machu Picchu was a royal estate or religious sanctuary for the Incan emperor Pachacuti.

Student: Why didn't they just build it on the ground?

Teacher: They thought it would be more likely to be seen from the outside.

Student: Can we just move on?

Teacher: No, it's important to understand the historical significance of Machu Picchu.

Student: I don't care. Just tell me what I need to know for the test.

Teacher: I'm trying to help you understand the material, not just memorize it.

Student: I don't need your help. Just give me the notes.

Teacher: [end of conversation]

5. Teacher: Today, we're going to talk about foreshadowing in literature.

Student: I'm not really into reading. Is there a way we could learn about it in a more creative way?

Teacher: No, I'm afraid not. Foreshadowing is an important concept that you need to understand.

Student: But I learn better through storytelling or games.

Teacher: That's too bad. You need to learn to focus on the material, even if it's not presented in your preferred style.

Student: Maybe we could act out a scene where there's foreshadowing?

Teacher: That would be a waste of time. We need to cover the key points of foreshadowing.

Student: I'm not sure I'm going to understand it this way.

Teacher: You will if you pay attention and ask questions.

Student: Can you at least give me some examples of foreshadowing?

Teacher: Sure. In 'Romeo and Juliet,' the prologue foreshadows the tragic end of the two lovers.

Student: That makes sense. How does foreshadowing help the reader?

Teacher: It builds suspense and keeps the reader engaged.

Student: Okay, I think I'm starting to get it.

Teacher: That's great. I'm glad you're understanding.

Student: Thanks for working with me. You're welcome. [end of conversation]

# D  Hyperparameters and additional experimental results

## D.1  Hyperparameters

For both RLHF-based and preference-based algorithms we conducted a sweep over the KL regularization coefficient $\alpha \in \{0.0025, 0.005, 0.01, 0.02, 0.05, 0.1\}$. For preference-based algorithm we also conducted a sweep over the mixing coefficient $\tau \in \{0, 0.0375, 0.0625, 0.125\}$. All models are trained for 50000 steps.

Table 4: Hyperparameters of all multi-turn algorithms.

| Hyperparameter | RLHF | MTPO | MTPO-$\tau$ |
|---|---|---|---|
| # generations per context | 1 | 2 | 2 |
| # updates per context | 1 | 2 | 2 |
| KL regularization coefficient $\alpha$ | 0.01 | 0.005 | 0.0025 |
| mixing coefficient $\tau$ | 0 | 0 | 0.0375 |
| batch size | 16 | 16 | 16 |
| GAE coefficient $\lambda$ | 0.95 | 0.95 | 0.95 |
| policy learning delay | 1000 | 1000 | 1000 |
| optimizer | AdaFactor | AdaFactor | AdaFactor |
| optimizer decay | 0.8 | 0.8 | 0.8 |
| policy learning rate | 4e-5 | 4e-5 | 4e-5 |
| value learning rate | 4e-5 | 4e-5 | 4e-5 |
| value initialization | pretrained checkpoint | pretrained checkpoint | pretrained checkpoint |

## D.2  Additional experimental results

Table (full version of Table 1): Side-by-side evaluation for Education Dialogue using Flan-T5 XL as the prompted preference model. Each entry is the average preference of 1,600 conversations generated with row method $y$, over ones generated with column method $y'$. We evaluate each method using 3 different seeds, compute $3 \times 3$ comparisons matrix and report the mean (together with the standard deviation).

| | SL | Single-turn-reward | | Single-turn-value | | Multi-turn | | |
|---|---|---|---|---|---|---|---|---|
| | SFT | RLHF | Nash | RLHF | Nash | RLHF | MTPO | MTPO-$\tau$ |
| SFT | – | 0.164 (.015) | 0.347 (.017) | 0.197 (.012) | 0.324 (.015) | 0.212 (.007) | 0.091 (.014) | 0.093 (.02) |
| RLHF-reward | 0.836 (.015) | – | 0.628 (.015) | 0.515 (.016) | 0.654 (.016) | 0.399 (.06) | 0.392 (.029) | 0.354 (.017) |
| Nash-reward | 0.653 (.017) | 0.372 (.015) | – | 0.411 (.02) | 0.51 (.021) | 0.328 (.077) | 0.281 (.02) | 0.242 (.019) |
| RLHF-value | 0.803 (.012) | 0.485 (.016) | 0.589 (.02) | – | 0.568 (.016) | 0.408 (.053) | 0.396 (.015) | 0.366 (.026) |
| Nash-value | 0.676 (.015) | 0.346 (.016) | 0.49 (.021) | 0.432 (.016) | – | 0.45 (.073) | 0.298 (.025) | 0.27 (.026) |
| RLHF-multi | 0.788 (.007) | 0.601 (.06) | 0.672 (.077) | 0.592 (.053) | 0.55 (.073) | – | 0.433 (.053) | 0.412 (.031) |
| MTPO | 0.909 (.014) | 0.608 (.029) | 0.719 (.02) | 0.604 (.015) | 0.702 (.025) | 0.567 (.053) | – | 0.439 (.034) |
| MTPO-$\tau$ | **0.907** (.02) | **0.646** (.017) | **0.758** (.019) | **0.634** (.026) | **0.73** (.026) | **0.588** (.031) | **0.561** (.034) | – |

# E Mirror descent policy optimization for regularized adversarial MDPs

## E.1 Model

We start by defining the regularized adversarial MDP model.

Consider a setting where the agent interacts with an MDP model for $T$ episodes, such that, in each episode $t \in [T]$, the agent performs $H$ steps in the MDP (from horizon $h = 1$ up to horizon $h = H + 1$) In short, an adversarial MDP is a generalization of this standard episodic MDP setting to the scenario where the reward function is different in every episode. This model was extensively studied in recent years (see, e.g., Even-Dar et al. [2009], Rosenberg and Mansour [2019b,a], Shani et al. [2020b]). We consider a slightly different definition which is more focused on our setting.

**Definition E.1** (Adversarial MDP). A finite-horizon adversarial MDP $\mathcal{M}$ is defined by a tuple $(\mathcal{X}, \mathcal{Y}, H, x_1, p, \{r^t\}_{t=1}^T)$ where $\mathcal{X}$ is the state space, $\mathcal{Y}$ is the action space, $H$ is the horizon, $x_1 \in \mathcal{X}_1$ is the initial state, $p : \mathcal{X} \times \mathcal{Y} \to \Delta_{\mathcal{X}}$ is the transition function, and $r^t : \mathcal{X}_{H+1} \to [0,1]$ is the reward function in episode $t$.

An interaction between the agent and the adversarial MDP environment proceeds in $T$ episodes, and each episode $t \in [T]$ proceeds in $H$ steps. The agent begins in an initial state $x_1^t = x_1$. In step $h \in [H]$, the agent observes the current state $x_h^t \in \mathcal{X}$, picks an action $y_h^t \in \mathcal{Y}$ and transitions to the next state $x_{h+1}^t$ sampled from the transition function $p(\cdot \mid x_h^t, y_h^t)$. At the end of the interaction, the agent arrives in a final state $x_{H+1}^t$ and observes the reward $r^t(x_{H+1}^t)$. For simplicity, we assume that the state space can be decomposed into $H + 1$ disjoint subsets $\mathcal{X} = \biguplus_{h=1}^{H+1} X_h$ such that, in step $h$ of the interaction, the agent is in some state $x_h \in \mathcal{X}_h$.

Now, we define the value function in an adversarial MDP, i.e., the expected reward of a policy when interacting with the MDP.

**Definition E.2** (Value function). Let $\mathcal{M}$ be an adversarial MDP and $\pi : \mathcal{X} \to \Delta_{\mathcal{Y}}$ be a policy. The value function $V^{\pi,t} : \mathcal{X} \to \mathbb{R}$ of policy $\pi$ in episode $t$ is defined as $V^{\pi,t}(x_h) = \mathbb{E}_{\pi,p}[r^t(x_{H+1}) \mid x_h]$ for every $h \in [H]$ and $x_h \in \mathcal{X}_h$. Similarly, the Q-function $Q^{\pi,t} : \mathcal{X} \times \mathcal{Y} \to \mathbb{R}$ is defined $Q^{\pi,t}(x_h, y_h) = \mathbb{E}_{\pi,p}[r^t(x_{H+1}) \mid x_h, y_h]$.

Next, we consider a regularized version of the adversarial MDP model. Regularized MDPs were also studied recently (see, e.g., Geist et al. [2019], Shani et al. [2020a]). The following definition presents the regularized value with respect to some reference policy $\mu$.

**Definition E.3** (Regularized value function). Let $\mu$ be a reference policy and $\alpha > 0$ be a regularization coefficient. The regularized value function and Q-function of policy $\pi$ in episode $t$ are defined as

$$V_\alpha^{\pi,t}(x_h) = \mathbb{E}_{\pi,p}\left[r^t(x_{H+1}) - \sum_{h'=h}^H \mathrm{KL}(\pi\|\mu)[x_h] \mid x_h\right]$$

$$Q_\alpha^{\pi,t}(x_h, y_h) = \mathbb{E}_{\pi,p}\left[r^t(x_{H+1}) - \sum_{h'=h}^H \mathrm{KL}(\pi\|\mu)[x_h] \mid x_h, y_h\right].$$

We now present a 1-step recursive formula and a value difference lemma for the regularized value function.

**Lemma E.4** (Regularized value function recursive relation). *Let $\pi$ be a policy. For every $x_{H+1} \in \mathcal{X}_{H+1}$, it holds that $V_\alpha^{\pi,t}(x_{H+1}) = r^t(x_{H+1})$. Furthermore, for every $h = 1, \ldots, H$ and $(x_h, y_h) \in \mathcal{X}_h \times \mathcal{Y}$, the following recursive relations hold:*

$$V_\alpha^{\pi,t}(x_h) = \sum_{y_h \in \mathcal{Y}} \pi(y_h \mid x_h) Q_\alpha^{\pi,t}(x_h, y_h)$$

$$Q_\alpha^{\pi,t}(x_h, y_h) = \sum_{x_{h+1} \in \mathcal{X}_{h+1}} p(x_{h+1} \mid x_h, y_h) V_\alpha^{\pi,t}(x_{h+1}) - \alpha \mathrm{KL}(\pi\|\mu)[x_h].$$

*Proof.* We prove the claim by backwards induction on $h$. The base case $h = H + 1$ follows by definition of the value function and the adversarial MDP. Assuming that the claim holds for $h + 1$, we

have that:

$$V_\alpha^{\pi,t}(x_h) = \mathbb{E}_{\pi,p}\left[r^t(x_{H+1}) - \alpha \sum_{h'=h}^{H} \mathrm{KL}(\pi\|\mu)[x_{h'}] \mid x_h\right]$$

$$= \sum_{x\in\mathcal{X}_{H+1}} \Pr_{\pi,p}[x_{H+1} = x \mid x_h]r^t(x) - \alpha \sum_{h'=h}^{H} \sum_{x\in\mathcal{X}_{h'}} \Pr_{\pi,p}[x_{h'} = x \mid x_h]\mathrm{KL}(\pi\|\mu)[x]$$

$$= \sum_{y_h\in\mathcal{Y}} \pi(y_h \mid x_h) \sum_{x\in\mathcal{X}_{H+1}} \Pr_{\pi,p}[x_{H+1} = x \mid x_h, y_h]r^t(x)$$

$$\qquad - \alpha \sum_{y_h\in\mathcal{Y}} \pi(y_h \mid x_h) \sum_{h'=h}^{H} \sum_{x\in\mathcal{X}_{h'}} \Pr_{\pi,p}[x_{h'} = x \mid x_h, y_h]\mathrm{KL}(\pi\|\mu)[x]$$

$$= \sum_{y_h\in\mathcal{Y}} \pi(y_h \mid x_h)\mathbb{E}_{\pi,p}\left[r^t(x_{H+1}) - \alpha \sum_{h'=h}^{H} \mathrm{KL}(\pi\|\mu)[x_{h'}] \mid x_h, y_h\right]$$

$$= \sum_{y_h\in\mathcal{Y}} \pi(y_h \mid x_h)Q_\alpha^{\pi,t}(x_h, y_h).$$

Moreover,

$$Q_\alpha^{\pi,t}(x_h, y_h) = \mathbb{E}_{\pi,p}\left[r^t(x_{H+1}) - \alpha \sum_{h'=h}^{H} \mathrm{KL}(\pi\|\mu)[x_{h'}] \mid x_h, y_h\right]$$

$$= \sum_{x\in\mathcal{X}_{H+1}} \Pr_{\pi,p}[x_{H+1} = x \mid x_h, y_h]r^t(x)$$

$$\qquad - \alpha \sum_{h'=h}^{H} \sum_{x\in\mathcal{X}_{h'}} \Pr_{\pi,p}[x_{h'} = x \mid x_h, y_h]\mathrm{KL}(\pi\|\mu)[x]$$

$$= \sum_{x_{h+1}\in\mathcal{X}_{h+1}} p(x_{h+1} \mid x_h, y_h) \sum_{x\in\mathcal{X}_{H+1}} \Pr_{\pi,p}[x_{H+1} = x \mid x_{h+1}]r^t(x)$$

$$\qquad - \alpha \sum_{x_{h+1}\in\mathcal{X}_{h+1}} p(x_{h+1} \mid x_h, y_h) \sum_{h'=h+1}^{H} \sum_{x\in\mathcal{X}_{h'}} \Pr_{\pi,p}[x_{h'} = x \mid x_{h+1}]\mathrm{KL}(\pi\|\mu)[x]$$

$$\qquad - \alpha\mathrm{KL}(\pi\|\mu)[x_h]$$

$$= \sum_{x_{h+1}\in\mathcal{X}_{h+1}} p(x_{h+1} \mid x_h, y_h)\mathbb{E}_{\pi,p}\left[r^t(x_{H+1}) - \alpha \sum_{h'=h+1}^{H} \mathrm{KL}(\pi\|\mu)[x_{h'}] \mid x_{h+1}\right]$$

$$\qquad - \alpha\mathrm{KL}(\pi\|\mu)[x_h]$$

$$= \sum_{x_{h+1}\in\mathcal{X}_{h+1}} p(x_{h+1} \mid x_h, y_h)V_\alpha^{\pi,\pi'}(x_{h+1}) - \alpha\mathrm{KL}(\pi\|\mu)[x_h]. \qquad \square$$

**Lemma E.5** (Regularized Value Difference Lemma). *Let $\pi, \pi'$ be two policies. Then,*

$$V_\alpha^{\pi,t}(x_1) - V_\alpha^{\pi',t}(x_1) = \mathbb{E}_{\pi',p}\left[\sum_{h=1}^{H} \langle \pi - \pi', Q_\alpha^{\pi,t}\rangle[x_h] + \alpha\mathrm{KL}(\pi'\|\mu)[x_h] - \alpha\mathrm{KL}(\pi\|\mu)[x_h]\right].$$

*Proof.* Let $x_h \in \mathcal{X}_h$. First, by Lemma E.5,

$$V_\alpha^{\pi,t}(x_h) - V_\alpha^{\pi',t}(x_h) =$$

$$= \langle \pi, Q_\alpha^{\pi,t}\rangle[x_h] - \langle \pi', Q_\alpha^{\pi',t}\rangle[x_h]$$

$$= \langle \pi - \pi', Q_\alpha^{\pi,t}\rangle[x_h] + \langle \pi', Q_\alpha^{\pi,t} - Q_\alpha^{\pi',t}\rangle[x_h]$$

$$= \langle \pi - \pi', Q_\alpha^{\pi,t}\rangle[x_h]$$

$$\qquad + \alpha\mathrm{KL}(\pi'\|\mu)[x_h] - \alpha\mathrm{KL}(\pi\|\mu)[x_h]$$

$$\qquad + \sum_{y_h\in\mathcal{Y}} \sum_{x_{h+1}\in\mathcal{X}_{h+1}} \pi'(y_h \mid x_h)p(x_{h+1} \mid x_h, y_h)\left(V_\alpha^{\pi,t}(x_{h+1}) - V_\alpha^{\pi',t}(x_{h+1})\right).$$

Note that for $h = H + 1$ for any $\pi, x_{H+1}$ we have $V_\alpha^{\pi,t}(x_{H+1}) = r^t(x_{H+1})$. By recursively unrolling the above relation, we get

$$V_\alpha^{\pi,t}(x_h) - V_\alpha^{\pi',t}(x_h) = \mathbb{E}_{\pi',p}\left[ \sum_{h'=h}^{H} \left\langle \pi - \pi', Q_\alpha^{\pi,t} \right\rangle [x_{h'}] \mid x_h \right]$$
$$+ \alpha \mathbb{E}_{\pi',p}\left[ \sum_{h'=h}^{H} \mathrm{KL}(\pi'\|\mu)[x_{h'}] - \mathrm{KL}(\pi\|\mu)[x_{h'}] \mid x_h \right].$$

Taking the expectation over the initial state finishes the proof. $\square$

## E.2 Algorithm 1: mirror descent policy optimization

We define the following mirror descent policy optimization algorithm. In the first episode the algorithm plays the reference policy, i.e., $\pi_1 = \mu$. Then, its update rule for iteration $(t + 1)$ is as follows:

$$\pi_{t+1}(\cdot \mid x_h) = \arg\max_\pi \eta_t \left\langle \pi, Q_\alpha^{\pi_t,t} \right\rangle [x_h] - \alpha\eta_t \mathrm{KL}(\pi\|\mu)[x_h] - (1 - \alpha\eta_t)\mathrm{KL}(\pi\|\pi_t)[x_h], \quad (3)$$

where $\eta_t$ is a learning rate. The solution can also be made explicit in the following form:

$$\pi_{t+1}(y_h \mid x_h) \propto \mu(y_h \mid x_h)^{\alpha\eta_t} \pi_t(y_h \mid x_h)^{1-\alpha\eta_t} e^{\eta_t Q_\alpha^{\pi_t,t}(x_h,y_h)}. \quad (4)$$

To show this, note that by the definition of the KL, we can write this update rule differently:

$$\pi_{t+1}(\cdot \mid x_h) = \arg\max_\pi \eta_t \sum_{y_h \in \mathcal{Y}} \pi(y_h \mid x_h)\left( Q_\alpha^{\pi_t,t}(x_h, y_h) - \alpha\log\frac{\pi_t(y_h \mid x_h)}{\mu(y_h \mid x_h)} \right) - \mathrm{KL}(\pi\|\pi_t)[x_h].$$

This is exactly the MD step for policy optimization [Orabona, 2019, Shani et al., 2020a]. Thus, the solution in its explicit form is:

$$\pi_{t+1}(y_h \mid x_h) \propto \pi_t(y_h \mid x_h) e^{\eta_t \left( Q_\alpha^{\pi_t,t}(x_h,y_h) - \alpha\log\frac{\pi_t(y_h|x_h)}{\mu(y_h|x_h)} \right)}.$$

We recover Equation (4) by noticing that $\exp\left( -\alpha\eta_t \log\frac{\pi_t(y_h|x_h)}{\mu(y_h|x_h)} \right) = \pi_t(y_h \mid x_h)^{-\alpha\eta_t} \mu(y_h \mid x_h)^{\alpha\eta_t}$.

## E.3 Analysis of algorithm 1

**Lemma E.6** (Fundamental inequality of mirror descent policy optimization for regularized adversarial MDPs). *The following holds when running mirror descent policy optimization (Equation (3)) in a regularized adversarial MDP, for every policy $\pi$ and every episode $t$,*

$$\mathrm{KL}_p(\pi\|\pi_{t+1}) \leq (1 - \eta_t\alpha)\mathrm{KL}_p(\pi\|\pi_t) + 2\eta_t^2 \mathbb{E}_{\pi,p}\left[ \sum_{h=1}^{H} \left\| Q_\alpha^{\pi_t,t}(x_h,\cdot) - \alpha\log\frac{\pi_t(\cdot \mid x_h)}{\mu(\cdot \mid x_h)} \right\|_\infty^2 \right]$$
$$+ \eta_t\left( V_\alpha^{\pi_t,t}(x_1) - V_\alpha^{\pi,t}(x_1) \right).$$

*Proof.* Fix a state $x_h \in \mathcal{X}_h$. We start by applying Munos et al. [2023, Lemma 2] with $\pi^+ = \pi_{t+1}$, $\pi^- = \pi_t$ and the vector $\delta(y) = \eta_t\left( Q_\alpha^{\pi_t,t}(x_h, y) - \alpha\log\frac{\pi_t(y|x_h)}{\mu(y|x_h)} \right)$. This implies that for any policy $\pi$,

$$\mathrm{KL}(\pi\|\pi_{t+1})[x_h] \leq \mathrm{KL}(\pi\|\pi_t)[x_h] + 2\eta_t^2 \left\| Q_\alpha^{\pi_t,t}(x_h,\cdot) - \alpha\log\frac{\pi_t(\cdot \mid x_h)}{\mu(\cdot \mid x_h)} \right\|_\infty^2$$
$$+ \eta_t\left\langle \pi_t(\cdot \mid x_h) - \pi(\cdot \mid x_h), Q_\alpha^{\pi_t,t}(x_h,\cdot) - \alpha\log\frac{\pi_t(\cdot \mid x_h)}{\mu(\cdot \mid x_h)} \right\rangle.$$

Next, we plug this into Lemma 3.1 to obtain

$$
\begin{aligned}
\mathrm{KL}_p(\pi\|\pi_{t+1}) = {}& \mathbb{E}_{\pi,p}\left[\sum_{h=1}^{H} \mathrm{KL}(\pi\|\pi_{t+1})[x_h]\right] \\
\leq {}& \mathbb{E}_{\pi,p}\left[\sum_{h=1}^{H} \mathrm{KL}(\pi\|\pi_t)[x_h]\right] \\
& + 2\eta_t^2 \mathbb{E}_{\pi,p}\left[\sum_{h=1}^{H} \left\|Q_\alpha^{\pi_t,t}(x_h,\cdot) - \alpha \log \frac{\pi_t(\cdot\mid x_h)}{\mu(\cdot\mid x_h)}\right\|_\infty^2\right] \\
& + \eta_t \mathbb{E}_{\pi,p}\left[\sum_{h=1}^{H} \left\langle \pi_t(\cdot\mid x_h) - \pi(\cdot\mid x_h), Q_\alpha^{\pi_t,t}(x_h,\cdot) - \alpha \log \frac{\pi_t(\cdot\mid x_h)}{\mu(\cdot\mid x_h)}\right\rangle\right].
\end{aligned}
$$

Note that

$$
\begin{aligned}
\left\langle \pi_t(\cdot\mid x_h) - \pi(\cdot\mid x_h), \log \frac{\pi_t(\cdot\mid x_h)}{\mu(\cdot\mid x_h)}\right\rangle = {} \\
= {}& \mathrm{KL}(\pi_t\|\mu)[x_h] - \left\langle \pi(\cdot\mid x_h), \log \frac{\pi_t(\cdot\mid x_h)}{\mu(\cdot\mid x_h)}\right\rangle \\
= {}& \mathrm{KL}(\pi_t\|\mu)[x_h] - \left\langle \pi(\cdot\mid x_h), \log \frac{\pi_t(\cdot\mid x_h)}{\mu(\cdot\mid x_h)}\right\rangle + \mathrm{KL}(\pi\|\mu)[x_h] - \mathrm{KL}(\pi\|\mu)[x_h] \\
= {}& \mathrm{KL}(\pi_t\|\mu)[x_h] + \mathrm{KL}(\pi\|\pi_t)[x_h] - \mathrm{KL}(\pi\|\mu)[x_h],
\end{aligned}
$$

where the last relation follows simply because:

$$
\begin{aligned}
\mathrm{KL}(\pi\|\mu)[x_h] - \left\langle \pi(\cdot\mid x_h), \log \frac{\pi_t(\cdot\mid x_h)}{\mu(\cdot\mid x_h)}\right\rangle = {} \\
= {}& \sum_{y_h\in\mathcal{Y}} \pi(y_h\mid x_h)\left(\log \frac{\pi(y_h\mid x_h)}{\mu(y_h\mid x_h)} - \log \frac{\pi_t(y_h\mid x_h)}{\mu(y_h\mid x_h)}\right) \\
= {}& \sum_{y_h\in\mathcal{Y}} \pi(y_h\mid x_h)\log \frac{\pi(y_h\mid x_h)}{\pi_t(y_h\mid x_h)} \\
= {}& \mathrm{KL}(\pi\|\pi_t)[x_h].
\end{aligned}
$$

Thus, we get

$$
\begin{aligned}
\mathrm{KL}_p(\pi\|\pi_{t+1}) &\leq \mathbb{E}_{\pi,p}\left[\sum_{h=1}^{H}\mathrm{KL}(\pi\|\pi_t)[x_h]\right]\\
&\quad + 2\eta_t^2\mathbb{E}_{\pi,p}\left[\sum_{h=1}^{H}\left\|Q_\alpha^{\pi_t,t}(x_h,\cdot)-\alpha\log\frac{\pi_t(\cdot\mid x_h)}{\mu(\cdot\mid x_h)}\right\|_\infty^2\right]\\
&\quad + \eta_t\mathbb{E}_{\pi,p}\left[\sum_{h=1}^{H}\langle\pi_t-\pi,Q_\alpha^{\pi_t,t}\rangle[x_h]\right]\\
&\quad - \eta_t\alpha\mathbb{E}_{\pi,p}\left[\sum_{h=1}^{H}\mathrm{KL}(\pi_t\|\mu)[x_h]+\mathrm{KL}(\pi\|\pi_t)[x_h]-\mathrm{KL}(\pi\|\mu)[x_h]\right]\\
&= (1-\eta_t\alpha)\mathbb{E}_{\pi,p}\left[\sum_{h=1}^{H}\mathrm{KL}(\pi\|\pi_t)[x_h]\right]\\
&\quad + 2\eta_t^2\mathbb{E}_{\pi,p}\left[\sum_{h=1}^{H}\left\|Q_\alpha^{\pi_t,t}(x_h,\cdot)-\alpha\log\frac{\pi_t(\cdot\mid x_h)}{\mu(\cdot\mid x_h)}\right\|_\infty^2\right]\\
&\quad + \eta_t\mathbb{E}_{\pi,p}\left[\sum_{h=1}^{H}\langle\pi_t-\pi,Q_\alpha^{\pi_t,t}\rangle[x_h]+\alpha\mathrm{KL}(\pi\|\mu)[x_h]-\alpha\mathrm{KL}(\pi_t\|\mu)[x_h]\right]\\
&= (1-\eta_t\alpha)\mathrm{KL}_p(\pi\|\pi_t) + 2\eta_t^2\mathbb{E}_{\pi,p}\left[\sum_{h=1}^{H}\left\|Q_\alpha^{\pi_t,t}(x_h,\cdot)-\alpha\log\frac{\pi_t(\cdot\mid x_h)}{\mu(\cdot\mid x_h)}\right\|_\infty^2\right]\\
&\quad + \eta_t\left(V_\alpha^{\pi_t,t}(x_1)-V_\alpha^{\pi,t}(x_1)\right),
\end{aligned}
$$

where the third relation is by Lemmas 3.1 and E.5. $\qquad\square$

## E.4  Algorithm 2: mixture mirror descent policy optimization

Define the mixture policy in iteration $t$ as:

$$
\pi_t^\alpha(y\mid x) = \frac{\pi_t(y\mid x)^{1-\eta_t\alpha}\mu(y\mid x)^{\eta_t\alpha}}{\sum_{y'\in\mathcal{Y}}\pi_t(y'\mid x)^{1-\eta_t\alpha}\mu(y'\mid x)^{\eta_t\alpha}}.
$$

We now define the following mixture mirror descent policy optimization algorithm. In the first episode the algorithm plays the reference policy, i.e., $\pi_1=\mu$. Then, its update rule for iteration $(t+1)$ is as follows:

$$
\pi_{t+1}(\cdot\mid x_h) = \arg\max_\pi \eta_t\sum_{y_h\in\mathcal{Y}}\pi(y_h\mid x_h)Q_\alpha^{\pi_t^\alpha,t}(x_h,y_h)-\mathrm{KL}(\pi\|\pi_t^\alpha)[x_h]. \tag{5}
$$

The solution can be made explicit:

$$
\pi_{t+1}(y_h\mid x_h) \propto \pi_t^\alpha(y_h\mid x_h)e^{\eta_t Q_\alpha^{\pi_t^\alpha,t}(x_h,y_h)}.
$$

## E.5  Analysis of algorithm 2

**Lemma E.7** (Fundamental inequality of mixture mirror descent policy optimization for regularized adversarial MDPs)**.** *The following holds when running mixture mirror descent policy optimization (Equation (5)) in a regularized adversarial MDP, for every policy $\pi$ and every episode $t$,*

$$
\begin{aligned}
\mathrm{KL}_p(\pi\|\pi_{t+1}) &\leq (1-\eta_t\alpha)\mathrm{KL}_p(\pi\|\pi_t) + 2\eta_t^2\mathbb{E}_{\pi,p}\left[\sum_{h=1}^{H}\left\|Q_\alpha^{\pi_t^\alpha,t}(x_h,\cdot)\right\|_\infty^2\right]\\
&\quad + \eta_t\left(V_\alpha^{\pi_t^\alpha,t}(x_1)-V_\alpha^{\pi,t}(x_1)\right).
\end{aligned}
$$

*Proof.* Fix a state $x_h\in\mathcal{X}_h$. We start by applying Munos et al. [2023, Lemma 2] with $\pi^+=\pi_{t+1}$, $\pi^-=\pi_t^\alpha$ and the vector $\delta(y)=\eta_t Q_\alpha^{\pi_t^\alpha,t}(x_h,y)$. This implies that for any policy $\pi$,

$$
\mathrm{KL}(\pi\|\pi_{t+1})[x_h] \leq \mathrm{KL}(\pi\|\pi_t^\alpha)[x_h] + \eta_t\langle\pi_t^\alpha-\pi,Q_\alpha^{\pi_t^\alpha,t}\rangle[x_h] + 2\eta_t^2\left\|Q_\alpha^{\pi_t^\alpha,t}(x_h,\cdot)\right\|_\infty^2.
$$

Next, we plug this into Lemma 3.1 to obtain

$$
\begin{aligned}
\mathrm{KL}_p(\pi\|\pi_{t+1}) = {} & \mathbb{E}_{\pi,p}\left[\sum_{h=1}^{H}\mathrm{KL}(\pi\|\pi_{t+1})[x_h]\right]\\
\leq {} & \mathbb{E}_{\pi,p}\left[\sum_{h=1}^{H}\mathrm{KL}(\pi\|\pi_t^\alpha)[x_h]\right]\\
& + \eta_t\mathbb{E}_{\pi,p}\left[\sum_{h=1}^{H}\left\langle\pi_t^\alpha-\pi,Q_\alpha^{\pi_t^\alpha,t}\right\rangle[x_h]\right] + 2\eta_t^2\mathbb{E}_{\pi,p}\left[\sum_{h=1}^{H}\left\|Q_\alpha^{\pi_t^\alpha,t}(x_h,\cdot)\right\|_\infty^2\right]\\
\leq {} & \mathbb{E}_{\pi,p}\left[\sum_{h=1}^{H}(1-\eta_t\alpha)\mathrm{KL}(\pi\|\pi_t)[x_h] + \eta_t\alpha\mathrm{KL}(\pi\|\mu)[x_h] - \eta_t\alpha\mathrm{KL}(\pi_t^\alpha\|\mu)[x_h]\right]\\
& + \eta_t\mathbb{E}_{\pi,p}\left[\sum_{h=1}^{H}\left\langle\pi_t^\alpha-\pi,Q_\alpha^{\pi_t^\alpha,t}\right\rangle[x_h]\right] + 2\eta_t^2\mathbb{E}_{\pi,p}\left[\sum_{h=1}^{H}\left\|Q_\alpha^{\pi_t^\alpha,t}(x_h,\cdot)\right\|_\infty^2\right]\\
= {} & (1-\eta_t\alpha)\mathbb{E}_{\pi,p}\left[\sum_{h=1}^{H}\mathrm{KL}(\pi\|\pi_t)[x_h]\right] + 2\eta_t^2\mathbb{E}_{\pi,p}\left[\sum_{h=1}^{H}\left\|Q_\alpha^{\pi_t^\alpha,t}(x_h,\cdot)\right\|_\infty^2\right]\\
& + \eta_t\mathbb{E}_{\pi,p}\left[\sum_{h=1}^{H}\left\langle\pi_t^\alpha-\pi,Q_\alpha^{\pi_t^\alpha,t}\right\rangle[x_h] + \alpha\mathrm{KL}(\pi\|\mu)[x_h] - \alpha\mathrm{KL}(\pi_t^\alpha\|\mu)[x_h]\right]\\
= {} & (1-\eta_t\alpha)\mathrm{KL}_p(\pi\|\pi_t) + 2\eta_t^2\mathbb{E}_{\pi,p}\left[\sum_{h=1}^{H}\left\|Q_\alpha^{\pi_t^\alpha,t}(x_h,\cdot)\right\|_\infty^2\right]\\
& + \eta_t\left(V_\alpha^{\pi_t^\alpha,t}(x_1) - V_\alpha^{\pi,t}(x_1)\right),
\end{aligned}
$$

where the second inequality is by Munos et al. [2023, Lemma 1], and the last relation are by Lemmas 3.1 and E.5. $\qquad\square$

### E.6  Bounding the Q-function

Define $\mu_{\min}$ as the minimal positive probability assigned by the reference policy $\mu$, i.e., $\mu_{\min} = \min_{(x,y)\in\mathcal{X}\times\mathcal{Y}:\mu(y|x)>0}\mu(y\mid x)$.

**Lemma E.8.** *For the choice $\eta_t = \frac{2}{\alpha(t+2)}$, in every iteration $t$ of mirror descent policy optimization it holds that*

$$
\alpha(H+1)\log\mu_{\min} \leq Q_\alpha^{\pi_t,t}(x,y) - \alpha\log\frac{\pi_t(y\mid x)}{\mu(y\mid x_h)} \leq 2 - \alpha H\log\mu_{\min} \qquad \forall(x,y)\in\mathcal{X}\times\mathcal{Y}.
$$

*Proof.* Follows directly from Lemmas E.9 and E.10. $\qquad\square$

**Lemma E.9.** *For every iteration $t$ of mirror descent policy optimization it holds that*

$$
\alpha H\log\mu_{\min} \leq Q_\alpha^{\pi_t,t}(x,y) \leq 1 \qquad \forall(x,y)\in\mathcal{X}\times\mathcal{Y}.
$$

*The same holds for $Q_\alpha^{\pi_t^\alpha,t}$ when running mixture mirror descent policy optimization.*

*Proof.* By the OMD optimization problem, $\pi_t(y\mid x)$ will not be positive unless $\mu(y\mid x)>0$ (the same holds for $\pi_t^\alpha$). Thus, for every $x\in\mathcal{X}$ we have

$$
\begin{aligned}
0 \leq \mathrm{KL}(\pi_t\|\mu)[x] &= \sum_{y\in\mathcal{Y}}\pi_t(y\mid x)\log\frac{\pi_t(y\mid x)}{\mu(y\mid x)}\\
&= \sum_{y\in\mathcal{Y}}\pi_t(y\mid x)\log\pi_t(y\mid x) - \sum_{y\in\mathcal{Y}}\pi_t(y\mid x)\log\mu(y\mid x)\\
&\leq \sum_{y\in\mathcal{Y}}\pi_t(y\mid x)\log\frac{1}{\mu(y\mid x)} \leq \log\frac{1}{\mu_{\min}}.
\end{aligned}
$$

Now, by definition, the Q-function is bounded from above by 1 and from below by $\alpha H\log\mu_{\min}$. $\qquad\square$

**Lemma E.10.** *For the choice $\eta_t = \frac{2}{\alpha(t+2)}$, in every iteration $t$ of mirror descent policy optimization it holds that*

$$H \log \mu_{\min} - \frac{1}{\alpha} \le \log \frac{\pi_t(y \mid x)}{\mu(y \mid x)} \le \log \frac{1}{\mu_{\min}} \qquad \forall (x, y) \in \mathcal{X} \times \mathcal{Y}.$$

*Proof.* For the upper bound, notice that

$$\log \frac{\pi_t(y \mid x)}{\mu(y \mid x)} = \log \pi_t(y \mid x) - \log \mu(y \mid x) \le \log \frac{1}{\mu(y \mid x)} \le \log \frac{1}{\mu_{\min}}.$$

For the lower bound, we repeat a similar analysis to that in Shani et al. [2020a, Lemma 25]. We start by bounding the partition function as follows

$$\log \sum_{y' \in \mathcal{Y}} \pi_t(y' \mid x) e^{\eta_t \left( Q_\alpha^{\pi_t, t}(x, y') - \alpha \log \frac{\pi_t(y' \mid x)}{\mu(y' \mid x)} \right)} \le \log \sum_{y' \in \mathcal{Y}} \pi_t(y' \mid x) e^{\eta_t \left( 1 - \alpha \log \frac{\pi_t(y' \mid x)}{\mu(y' \mid x)} \right)}$$

$$= \eta_t + \log \sum_{y' \in \mathcal{Y}} \pi_t(y' \mid x) \left( \frac{\mu(y' \mid x)}{\pi_t(y' \mid x)} \right)^{\eta_t \alpha}$$

$$\le \eta_t + \log \left( \sum_{y' \in \mathcal{Y}} \pi_t(y' \mid x) \frac{\mu(y' \mid x)}{\pi_t(y' \mid x)} \right)^{\eta_t \alpha} = \eta_t,$$

where the first inequality follows since $Q_\alpha^{\pi_t, t}(x, y) \le 1$, and the second uses Jensen inequality (with the fact that $\eta_t \alpha \le 1$). Now, by the update rule of $\pi_t$,

$$\log \frac{\pi_{t+1}(y \mid x)}{\mu(y \mid x)} = \log \frac{\pi_t(y \mid x)}{\mu(y \mid x)} + \eta_t Q_\alpha^{\pi_t, t}(x, y) - \eta_t \alpha \log \frac{\pi_t(y \mid x)}{\mu(y \mid x)}$$

$$- \log \sum_{y' \in \mathcal{Y}} \pi_t(y' \mid x) e^{\eta_t \left( Q_\alpha^{\pi_t, t}(x, y') - \alpha \log \frac{\pi_t(y' \mid x)}{\mu(y' \mid x)} \right)}$$

$$\ge (1 - \eta_t \alpha) \log \frac{\pi_t(y \mid x)}{\mu(y \mid x)} + \eta_t Q_\alpha^{\pi_t, t}(x, y) - \eta_t$$

$$\ge (1 - \eta_t \alpha) \log \frac{\pi_t(y \mid x)}{\mu(y \mid x)} + \eta_t (\alpha H \log \mu_{\min} - 1). \qquad \text{(Lemma E.9)}$$

To finish the proof plug in $\eta_t$ and unroll $t$ to 0. $\qquad \square$

