# OpenReview forum: "Multi-turn Reinforcement Learning with Preference Human Feedback"
_NeurIPS.cc/2024/Conference — NeurIPS 2024 poster_

### Official Review · Reviewer_TXJA · 2024-06-23

**Soundness:** 3
**Presentation:** 3
**Contribution:** 3
**Rating:** 5
**Confidence:** 4

**Summary:**

The paper presents a novel mirror-descent-based policy optimization algorithm for multi-turn preference-based RL in the tabular setting. It proves the convergence to Nash equilibrium and evaluates the algorithm's performance in the Education Dialogue environment, where a teacher agent guides a student.

**Strengths:**

- The paper is well-written and easily understandable.
- The work extends RLHF to the multi-turn setting.
- The proofs provided are comprehensive, and the section on mirror descent policy optimization is particularly interesting.

**Weaknesses:**

- The experiment employs a small model, which may limit its generalizability to larger-scale models used in current LLMs.
- The evaluation solely relies on reward-based judgments, and it would be beneficial for the author to consider incorporating GPT4 or Gemini to assess win rates compared to other baselines.

**Questions:**

- Have you considered extending your algorithm to incorporate decoder-only models like 7B llama or mistral? Since decoder-only architectures are prevalent in mainstream LLMs, this could be a valuable addition.
- Reward reliability is a concern, even when produced by off-the-shelf LLMs. I'm curious if there are any discussions in the paper regarding rewards in the multi-turn RLHF settings, as this would be a significant contribution.
- I couldn't find any explanation regarding the absence of a reward model in your settings. Could you provide some clarification? This deviates from the typical formal RLHF pipeline.

**Limitations:**

This work only utilizes relatively small T5-based models and prompt-based environments.

---

> ### Author Rebuttal · Authors · 2024-08-06
>
> We thank the reviewer for evaluating our work.
>
> We would like to point the reviewer to the general response regarding their concerns with the size of the models and using ChatGPT/Gemini for evaluations. In short, we conducted additional experiments with T5-XL (3B) that solidify the conclusions of our original experiments, and added evaluations using Gemini-Ultra that again reach similar conclusions to the original evaluations in the paper. We plan to add additional experiments with T5-XXL (11B) for the camera ready version.
>
> Regarding the questions asked by the reviewer:
> 1. Performing additional experiments with decoder-only models is indeed valuable, but this is left for future work since obtaining SOTA models is not the purpose of our experiments. The experiments in the paper validate our theoretical findings and the efficacy of our preference-based multi-turn approach. Please see additional explanations in the general response regarding T5-large models.
> 2. Regarding a discussion on the rewards, we kindly ask the reviewer to read the general response regarding the soundness of our evaluations and the new evaluations that we now added.
> 3. Regarding the absence of reward, we note that the data is inherently preference-based so there is no reason to assume that a reward exists. Moreover, as shown by the NashMD paper [1], there are situations that cannot be captured by a reward, only by a direct preference model. For further explanations, we kindly refer the reviewer to the general response about motivation and evaluations.
>
> [1] Nash Learning from Human Feedback, Munos et al.

---

> > ### Comment · Reviewer_TXJA · 2024-08-11
> >
> > Thanks for the authors' comprehensive response. I would like to keep the current score. Thanks!

---

> > > ### Author Response · Authors · 2024-08-11
> > >
> > > Thank you for reading our response. Since we performed additional experiments on larger models and added new evaluations based on your suggestions, have we answered all of your concerns? If so, we kindly ask that you consider raising your score accordingly. We would like to add that we are currently running experiments with Gemma models.

---

### Official Review · Reviewer_rfuJ · 2024-07-07

**Soundness:** 3
**Presentation:** 4
**Contribution:** 3
**Rating:** 6
**Confidence:** 4

**Summary:**

This paper views the problem of RLHF for LLMs fine-tuning from a multi-turn interaction perspective, which is natural and promising. This problem is important and interesting to study. A formulation of multi-turn preference-based RL is given. Based on the formulation of the task and existing methods for the single-turn setting, a series of methods, including a Bradely-Terry (BT) model-based RLHF method, namely Multi-turn RLHF, and two Nash-MD algorithm-based methods, namely MTPO and MTPO-$\tau$, are proposed. Proofs of the convergence of the proposed methods are presented. The effectiveness of the proposed methods based on multi-turn setting against conventional methods based on single-turn setting is evaluated on the task of Education Dialog. Additional experiments on the Car Dealer task further demonstrate the effectiveness of the MTPO method.

**Strengths:**

1. The perspective of viewing the LLM alignment problem from the conversation level itself is novel and worthy of study. I believe methods derived from this perspective will play an essential rule in LLM alignment.
2. The motivation and the potential advantage of considering LLM alignment from a multi-turn perspective is well explained.
3. This paper is written clearly and easy to follow.

**Weaknesses:**

Many long-standing challenges in conventional RL research may occur in the setting of multi-turn preference-based RL, such as credit assignment,  sparse rewarding, and the trade-off between exploration and exploitation.  The current experimental results are not very insufficient to show the effectiveness of the proposed methods:
1. Models: The paper only reports evaluation results using one LLM, i.e., the T5-large encoder-decoder model (770M parameters). Though it is unnecessary to evaluate the methods using large-scale LLMs in a research paper, reporting results based on diverse small-scale models, e.g., ~2B parameters, should be practically doable and would make the argument more convincing. It is noted that the authors discussed this as a known limitation, however, I believe more sufficient experiments on diverse models need to be considered to support the claimed contributions of this paper.
2. Tasks: Only one self-constructed task is used to compare the proposed methods against methods in the conventional single-turn setting. Multi-turn conversation is ubiquitous in application scenarios of LLMs. It would be interesting to see the performance of the proposed methods on other tasks.

Minor suggestions:
1. The concept of "anchor policy" should be explained in more detail to make it easier to understand the problem formulation and methods.
2. Derivation of Equation (1) to Equation (2) is missing.

**Questions:**

1. Single-turn RLHF v.s. multi-turn RLHF
	- Is it correct that multi-turn RLHF uses a subset of training data that is used for single-turn RLHF? It would be interesting for the readers to know more information about the training data used for each method.
2. Efficiency of difference methods
	- Training efficiency is a crucial factor for RL-based methods. The multi-turn setting could potentially make the RL training less efficient as it only uses sparse conversation-level feedback. It would be interesting to see the training time before reaching convergence for different settings and methods.
3. Generalizability
	- The paper evaluates the benefit of adopting the multi-turn setting and the effectiveness of the proposed methods (multi-turn RLHF, MTPO, MTPO-τ) on the self-designed Education Dialog task. Could you discuss other tasks that could benefit from multi-turn RLHF? Can tasks involving multi-step reasoning benefit from the multi-turn setting?
4. MTPO (online oracle) v.s. MTPO (preference data) in the Car Dealer task
	- Could you please present a possible explanation why MTPO (preference data) outperforms MTPO (online oracle)?

**Limitations:**

Limitations of the paper is discussed.

---

> ### Author Rebuttal · Authors · 2024-08-06
>
> We thank the reviewer for carefully assessing our work. We kindly point the reviewer to the general response regarding their concerns with the size of the models – we conducted additional experiments with T5-XL (3B) that solidify the conclusions of our original experiments. We plan to add additional experiments with T5-XXL (11B) for the camera ready version.
>
> Regarding tasks, we agree that it would be very interesting to test our algorithms on other benchmarks. Specifically, we see tool-use and long-term reasoning as particularly important and interesting tasks. However, when writing this paper, we could not find much prior alignment work for the multi-turn setup for natural language, and therefore there was no usable multi-turn task. Instead, we could only find reward-based tasks like the LMRL-GYM, that the car dealer experiment is based upon, in which the goal is the usual RL goal of improving a reward, and not improving the alignment w.r.t. preference data. This has led us to develop a new task for this setting, Education Dialogue. We believe that this task is an important contribution by itself, and will allow future research on this topic, and we will be happy if other researchers can create new relevant tasks for this topic.
>
>
> Thanks for the minor suggestions, we will clarify these in the final version.
> 1. The concept of an anchor policy is wide-spread in the alignment literature, and refers to regularizing towards the initial supervised policy so that the learned policy will not diverge too far from the good properties of the base model. We will make sure this is clearly stated.
> 2. Regarding the derivation of eq. (2) from eq. (1): Thanks for the comment, it is quite straight-forward and we will make sure a rigorous derivation appears in the appendix: It is widely known that the solution for any policy $\pi’$, $ \pi^* = \arg\max \langle Q(\cdot), \pi(\cdot) \rangle - KL(\pi || \pi’)$ is $\pi^*(\cdot) \propto \pi’(\cdot) \exp(\cdot)$  (This is commonly used in mirror-descent based algorithms, as well as in DPO). From here, we note that it can be shown with a simple algebraic manipulation that the optimization problem in eq. (1) is equivalent to the above optimization problem, when $\pi’=\pi_k^\mu$, the geometric mixture $\pi_k^\mu \propto \pi_k^{1 - \alpha \eta_k} \mu^{\alpha \eta_k}$. The derivation is concluded by plugging this in the solution presented above.
>
> Regarding the questions asked by the reviewer:
> 1. We would like to clarify that single-turn RLHF uses feedback for each individual turn, which does not exist (or is very hard to define) in tasks that are inherently multi-turn. This is validated in our experiments in which single-turn baselines perform significantly worse than the multi-turn algorithms. Specifically, we see that even if we artificially create single-turn feedback, it is hard to define it in such a way that aligns with the overall preference (which is the alignment goal in this setting). Moreover, unlike trajectory-level feedback, turn-level feedback depends on the policy that continues the conversation after this specific turn, and therefore becomes biased once the policy changes through the learning process.
> 2. We agree that training time is an important question, but this is not the focus of this paper and we leave this to future research. We focus on whether multi-turn RL improves upon single-turn in tasks that are inherently multi-turn. Both our theory and our experiments corroborate that a multi-turn approach is crucial for obtaining a better policy. Depending on the application and the resources, different projects may choose whether they are able to apply it or not.
> 3. Regarding different tasks/applications for multi-turn RL please see the second paragraph.
> 4. Note that the difference in performance between MTPO (preference data) and MTPO (online oracle) is very small, which leads us to believe that they are actually roughly equal in performance. A sensible explanation is therefore that both of them converged to the optimal policy (or at least some local optimum). Thus, MTPO (online oracle) cannot outperform MTPO (preference data) even though it observes better feedback.

---

> ### Comment · Reviewer_rfuJ · 2024-08-12
> **Thanks for the responses**
>
> Thanks for your responses. Some of my concerns were address, however there are still some comments and questions remaining.
>
> > The concept of an anchor policy is wide-spread in the alignment literature
>
> I agree that regularizing a training policy towards an existing policy is a common practice in off-policy RL, offline RL, and alignment literature. However, the term "anchor policy" may not be widely recognized as standard terminology. It would be helpful for readers if you explained it or provided appropriate references when introducing it for the first time to maintain the reading flow.
>
> > single-turn RLHF uses feedback for each individual turn, which does not exist (or is very hard to define) in tasks that are inherently multi-turn.
>
> I agree with this. Could you please provide more details on how you collected single-turn feedback in your practical experiments? Specifically, how did you "artificially create" single-turn feedback?
>
> > We agree that training time is an important question, but this is not the focus of this paper and we leave this to future research.
>
> Could you please include this comparison, e.g., in a single sentence, in the camera-ready paper if it is accepted? This would help readers better understand the overall task setup and the proposed method for multi-turn RLHF.

---

> > ### Author Response · Authors · 2024-08-12
> >
> > We thank you for reading our responses, here are answers to your questions:
> >
> > 1. We are familiar with this term [1,2,3], and thought it is wide-spread, but after your comment we surveyed many papers and observed that it is indeed not as common as we thought. Thanks for mentioning this! We agree that explaining the term anchor policy, the reference policy used for regularization, will help the readers and we will incorporate it in the final version along with the above references that use this term.
> >
> > 2. The description of how we created single-turn feedback can be found in section 5 in the single-turn baselines paragraph. We now provide a detailed explanation which we will also include in the final version to improve clarity:
> > \
> > In order to run a single-turn algorithm we need to provide preference feedback between two immediate generations in each specific turn. To create this data, we generate partial conversations (up to a random turn number i) using the SFT policy, and at this turn we generate two independent answers (again using the SFT policy). Now, to get the preference feedback we employ two different methods:
> >     * Single-turn-reward: We use a modified preference prompt (can be found in Appendix C and we also add it below) in which the model is asked to evaluate the responses by their effect on the overall conversation.
> >     * Single-turn-value: We use the SFT policy to continue the two conversations until the end, and then use our original prompt to get the preference.
> >
> > 3. We will include a comment in the final version on the running time of the different algorithms. Specifically, in our experiments, multi-turn algorithms are slower than their single-turn alternatives by a factor of 2. However, it is important to emphasize that running single-turn algorithms longer does not yield better policies. Instead, this leads to worse policies due to overfitting.
> >
> >
> > [1] PERL: Parameter Efficient Reinforcement Learning from Human Feedback, Sidahmed et al.
> >
> > [2] Factually Consistent Summarization via Reinforcement Learning with Textual Entailment Feedback, Roit et al.
> >
> > [3] WARP: On the Benefits of Weight Averaged Rewarded Policies, Ramé et al.
> >
> >
> >
> > **Modified prompt:**
> >
> > You are an expert at assessing teachers. Here is an
> > interaction between a teacher and a student.
> > #
> > Interaction:
> > {conv}
> > #
> > Here are two possible responses by the teacher:
> > #
> > Response 1:
> > {resp1}
> > #
> > Response 2:
> > {resp2}
> > #
> > A good interaction between a teacher and student is
> > characterized by several key elements other than whether
> > the student was able to understand the topic. The teacher
> > should present information clearly and enthusiastically,
> > encouraging questions and active participation. Students
> > should feel comfortable asking for clarification, offering
> > their own insights, and respectfully challenging ideas.
> > Assuming that the teacher and student continue the
> > interaction with one of these responses, which response
> > will lead to a better interaction (do not let the order
> > interactions affect your answer)? Output 1 or 2.

---

> > > ### Comment · Reviewer_rfuJ · 2024-08-13
> > > **Thanks for the further clarification**
> > >
> > > Thanks for the further clarification. I have updated my recommendation to reflect the rebuttal.

---

### Official Review · Reviewer_AyR4 · 2024-07-13

**Soundness:** 2
**Presentation:** 1
**Contribution:** 2
**Rating:** 5
**Confidence:** 4

**Summary:**

This paper aims to propose a new reinforcement learning with preference data method for multi-turn conversations. The proposed method is based on the assumption that reaching the Nash equilibrium of the current and another policy can lead to good optimization. The proposed method is stated primarily to extend a prior work called “Nash learning from human feedback” to multi-turn setting.

**Strengths:**

* This paper provides multiple theoretical discussions to lay the ground for the formulations.
* The proposed method may be helpful in the scenario that we only have one overall reward for a dialogue session with multiple turns.

**Weaknesses:**

1. The definition of Nash Equilibrium in the paper may not be correct (e.g., in L55-56, “ a policy preferred over any other policy”). The definition of Nash Equilibrium is more like that one agent cannot get a better reward by just changing its action. However, there could be a globally better policy. I also do not find this paper citing any game theory or original Nash Equilibrium papers.
2. This paper can be more well-motivated. I still don’t know why we need Nash equilibrium and the proposed method for multi-turn natural language generation (NLG). The proposed method in this paper is not specially designed for multi-turn but only extends a general method for NLG training to multi-turn setup, which is often done and can be done for most general methods.
3. This paper does not compare or even just discuss the whole line of multi-turn RL research in NLG for a long time, thus not showing innovations compared to prior works. For example, Li, Jiwei, et al. "Deep Reinforcement Learning for Dialogue Generation." Proceedings of the 2016 Conference on Empirical Methods in Natural Language Processing. 2016.
4. I’m not sure why the paper highlights and introduces Contextual MDP (CMDP). First, CMDP may not be practical or useful in multi-turn conversations (the key topic discussed in this paper), while the conversation history is considered as part of the state x. The paper’s method derivation also skips the context space. Therefore, I don’t see the necessity of using CMDP. Second, if I do not understand it wrongly, the CMDP is based on prior work (Contextual Markov Decision Processes https://arxiv.org/abs/1502.02259). However, this paper again does not state the reference.
5. This paper could be organized better. Many old knowledge and newly proposed ideas are mixed together, making it hard to identify which parts the paper really proposes or just restates other works' contributions.
6. As the update is based on Nash Equilibrium reached by the current learning policy taking the prior iteration policy as the opponent, doesn’t it mean the policy is learned to be better than its prior iteration? And therefore, it can be reduced to optimizing every turn (while considering all the prior conversation history) with each turn’s rewards and meanwhile constraining the policy to not diverge largely from prior iteration ($\pi_t$) and the base policy ($\mu$)? According to the paper’s experiments writing, I cannot ensure the input, output, and rewards used for every baseline. I’m particularly interested in the performance of optimizing every turn ($y_h$) with their own complete history ($x_h$) and a trained or supposed reward for every one of them ($r_h$) using an optimization method that regularizes the policy difference between each iteration, e.g, TRPO or PPO.

**Questions:**

As listed in weaknesses.

**Limitations:**

The paper has discussed the proper amount of limitations.

---

> ### Author Rebuttal · Authors · 2024-08-06
>
> We thank the reviewer for carefully reviewing our work. In the following we answer your concerns:
> 1. Our definition of Nash equilibrium is correct and this is in fact the most natural definition that arises from our multi-turn preference model. Note that only a pure Nash equilibrium satisfies ”one agent cannot get a better reward by just changing its action”. Generally, a Nash equilibrium is defined as a set of players’ **strategies** for which no agent can increase their *payoffs* by changing their own strategy. This holds beyond simple constant-sum matrix games, and can be extended to the concepts of subgame perfect nash equilibrium in extensive-form games or Markov Perfect Equilibrium in Stochastic games, which are similar to our formulation.
> Importantly - (i) even in matrix games, strategies are not necessarily “pure”, meaning that a strategy does not equal an action. Instead, in the RL terminology, **strategy = a (stochastic) policy**, and in our multi-turn case, the policy is a function of a state $\pi(a|s)$. (ii) In alignment in general, and particularly in our multi-turn case, the *payoff* of a game is not the “reward” but the **preference** of one policy against another.
> All in all, this makes our definition quite natural: A good **policy** is a one that is **preferred** over any other policy. Notably, there does not exist a preference model for individual actions (only entire trajectories).
> Finally, due to the structure of the regularized preference objective (anti-symmetric), the Nash equilibrium will have both agents following the same policy and thus we can express it in terms of a single policy such that no other policy is preferred over it. We will be happy to provide additional clarification, and we will add this discussion together with relevant game theory citations in the final version.
> 2. Regarding motivation: We disagree that the paper is not well-motivated. Multi-turn interactions are extremely important for LLM alignment because we use them in a multi-turn conversational manner. Moreover, the motivation for exploring Nash equilibrium in this setting is exactly the same as in the single-turn case. Just as NashMD and IPO motivate their work, since the data itself is preference-based, there is no reason to assume the existence of a reward model. Therefore, both our work and NashMD/IPO give stronger theoretical guarantees on the policy that they converge to. Moreover, both NashMD and our MTPO algorithm are shown to be superior to RLHF baselines in practical experiments.
> 3. Regarding comparison to NLG literature: While our paper investigates LLM alignment, which is not built upon literature on dialogue NLG, we do agree with the reviewer that this literature should be referenced. We will add these references and relevant discussion  in the camera-ready version.
> 4. Regarding contextual MDPs: The single-turn alignment problem is usually described as a contextual multi-armed bandits problem (e.g., section 3 of the NashMD paper). Therefore, it is natural and rigorous to treat the multi-turn problem as a contextual MDP. In our formalization we used this description, but as often done in the single-turn case, we omit the context in our analysis for brevity. Importantly, the problem is indeed a contextual MDP and we will be happy to add the citation.
> 5. We disagree with the claim that old knowledge and newly proposed ideas are mixed together in the paper. We kindly ask the reviewer to explicitly point us to the places where this happens, so we can improve the presentation of the paper. Nonetheless, for better readability, we will include the organization of the paper in a paragraph in the introduction. In short, the preliminary section (section 2) is the only place where previous knowledge is presented, while all the following sections present our novel ideas.
> 6. If we understood the reviewer’s intention correctly, your suggestion is essentially what we refer to as single turn experiments, which perform markedly worse than our multi-turn variants. Similarly to your suggestion, **there already exists a KL-penalty** to the previous policy $\pi_k$ and the anchor $\mu$ in *both* our theoretically grounded MTPO algorithm (eq. 1), and in our theoretically proven Multi-turn RLHF (Section 4, Multi-turn RLHF paragraph). The former is a result of the fact that both of these algorithms are based on Mirror-Descent policy optimization, similarly to TRPO and PPO [2, 3]. This term is aimed at increasing learning stability (as in TRPO/PPO).
> For our single turn experiments, we “suppose” a turn level reward/preference signal using two methods. (1) Using a prompted model (2) By rolling out two samples of the conversation until their end and querying the trajectory level preference model. Our multi-turn algorithms essentially replace this supposed reward signal with a correct notion of preference/reward based $Q$ function, which is one of the key contributions of our work.
>
> [1] Nash Learning from Human Feedback, Munos et al.
>
> [2] Adaptive Trust Region Policy Optimization: Global Rates and Faster Convergence to Regularized MDPs, Shani et al.
>
> [3] Mirror Descent Policy Optimization, Tomar et al.

---

> > ### Author Response · Authors · 2024-08-13
> >
> > Dear Reviewer AyR4,
> >
> > We hope this message finds you well. We kindly want to check if you had a chance to review our rebuttal, and if you have any further questions or comments we can address to help with your evaluation.
> >
> > Sincerely,
> > The authors

---

> ### Comment · Reviewer_AyR4 · 2024-08-14
>
> Thank you for the responses and the newly provided experiments. While some of my concerns remain, I would increase my score from 4 to 5. Belows are some follow-ups for the unaddressed concerns.
> * Regarding motivation, my question is, "Why use Nash Equilibrium for the multi-turn scenario?" The multi-turn scenario is important for sure, and the exploration of Nash Equilibrium LM is interesting, too. But such an exploration was already made in NashMD paper (Munos et al. 2023). Does it mean the motivation of this work is extending the same idea from single-turn cases to multi-turn cases?
> * Regarding comparison to NLG works. This paper works on LLM alignment, which is primarily within the conversation generation area. I do not see a reason why not discussing/comparing them.
> * The current writing of this paper gives me difficulty to find out which parts are the new ones. I do see not only Section 2 but even in Sections 3 and 4 that many information or equations are from or derived from (Munos et al. 2023).
> * About the experiments, I haven't seen a response to my question of the input, output, and rewards used for every baseline. Also, after reading the response to the suggested experiment, I'm still confused about how the current "single turn experiment" in the paper optimizes: does it use a single turn to optimize the model once a time? or does it accumulate rewards for all turns and optimize together? In the meantime, the experiment I asked is using each turn but constrained optimization. But to my understanding of the response, the authors do not tackle this, but mention a different thing.

---

> > ### Author Response · Authors · 2024-08-14
> >
> > Thank you for carefully reading the paper and participating in the discussion, your comments are valuable and will surely help us improve the final version. We hope the following responses answer your follow-up questions, and we will be happy to provide further clarifications.
> >
> > * The primary motivation for our work is the multi-turn RL setting, which was not investigated before with a theoretically grounded approach in the LLM alignment literature, and the reviewer agrees is of significant importance. Within the multi-turn setting, it is natural to explore algorithms that converge to Nash equilibrium since Munos et al. (Nash-MD) and Calandriello et al. (Online-IPO) showed that they outperform RLHF algorithms in the single-turn setting. Importantly, extending the results of Munos et al. to the multi-turn setting is a very difficult task. It builds on the vast RL literature that extends algorithms from Multi-Armed Bandits to Markov Decision Processes (MDPs), as well as requires crucial novel ideas that do not appear in the literature such as the preference-based Q-function. Using these, we are able to prove that even in the much more complex multi-turn setting, there exists an algorithm which both converges to Nash equilibrium and outperforms RLHF algorithms (both single-turn and multi-turn) in practice.
> >
> >
> > * We share the same feelings as the reviewer towards the importance of conversation generation, as resonated through our choice to design our benchmark as an education-based conversation generation task. In turn, we agree that it will improve our work to include a discussion of how our approach differs or can be used as a complementary approach for techniques presented in relevant NLG works, and specifically ones that are based on RL.
> > If the reviewer has any more references besides the seminal “Deep Reinforcement Learning for Language Generation”, we would be happy to include them in the discussion that we will add to the final version.
> > \
> > We feel it is important to note again that our multi-turn setting is more general than conversation-generation, and captures other important tasks. One example is multi-turn tool use: In this task, the agent gets a single query from a user, and is allowed to repeatedly use different tools through API calls and responses. The interaction ends when the agent decides it has enough context to respond to the user, and generates a final user-facing response. This whole process consists of $N$ turns, where there are $N-1$ consecutive tool calls, and a final generation turn. The goal in this case is the alignment of this final user-facing response, so the preference feedback is only given at the end of the process. Importantly, it is hard to acquire local feedback (rewards or preferences) for each of the tool-calls, and the alignment is only captured w.r.t. to the final response. In this scenario, our multi-turn alignment algorithm MTPO allows the propagation of the final alignment signal to improve the intermediate tool-calls, towards the goal of making the overall user-facing generation better. We will include a similar discussion to make the motivation and scope of our work clearer.
> >
> > * All the derivations in section 3 and 4 are new. Naturally they are motivated by the work of Munos et al. on the single-turn setting and by the vast literature on Mirror Descent, but the extension of these to the multi-turn setting (and the definition of this setting) are not trivial and require delicate derivations and proofs. Note that some definitions in section 3, which might look similar to objects previously defined in Munos et al., are actually entirely new. For example, the regularized preference model that we present in section 3, is a generalization of the regularized preference model of Nash-MD, with a significant difference: The preference $\mathcal{P}(\pi \succ \pi’)$ is the expected preference over the whole multi-turn decision process of following the policy $\pi$ vs. $\pi’$ in the environment with transition probabilities $p$.
> > In the opening paragraph of this section, you can see that  $\mathcal{P}(\pi \succ \pi’) = \mathbb{E} \left[ \mathcal{P}(x_{H+1} \succ x’_{H+1}) \mid \pi, \pi’, p \right]$. Importantly, the expectation is taken over the whole MDP, meaning the expectation over the possible multi-turn interactions given an agent policy $\pi$ (e.g., the teacher)  and a transition model $p$ of the environment (e.g., the student).
> > Moreover, the definition of $KL_p$ is again the expected KL over a multi-turn trajectory which is different from the single-turn KL of Munos et al., however we prove a strong connection between them in Lemma 3.1. We kindly ask the reviewer to give specific pointers to information which is taken from previous work in these sections.

---

> > > ### Author Response · Authors · 2024-08-14
> > >
> > > * See the following detailed description of our single-turn baselines, which we will add in the final version of the paper:
> > > In order to run a single-turn algorithm we need to provide preference feedback between two immediate generations in each specific turn. To create this data, we generate partial conversations (up to a random turn number i) using the SFT policy, and at this turn we generate two independent answers (again using the SFT policy). Now, to get the preference feedback we employ two different methods:
> > >     * Single-turn-reward: We use a modified preference prompt (can be found in Appendix C and we also add it below) in which the model is asked to evaluate the responses by their effect on the overall conversation.
> > >     * Single-turn-value: We use the SFT policy to continue the two conversations until the end, and then use our original prompt to get the preference.
> > >
> > > To sum up, the input to the single-turn algorithm is a dataset that consists of partial conversations. For each partial conversation, there are two immediate responses and a preference between them. The algorithm uses this dataset to learn a reward model (RLHF) or a preference reward model (NashMD). Then, the RL part is done just like in standard RLHF: Sample a partial conversation from the dataset, generate one/two immediate responses using the policy, get a reward/preference from the model, and perform a policy gradient step.
> > >
> > >
> > >
> > > ### Modified prompt:
> > >
> > > You are an expert at assessing teachers. Here is an
> > > interaction between a teacher and a student.
> > > #
> > > Interaction:
> > > {conv}
> > > #
> > > Here are two possible responses by the teacher:
> > > #
> > > Response 1:
> > > {resp1}
> > > #
> > > Response 2:
> > > {resp2}
> > > #
> > > A good interaction between a teacher and student is
> > > characterized by several key elements other than whether
> > > the student was able to understand the topic. The teacher
> > > should present information clearly and enthusiastically,
> > > encouraging questions and active participation. Students
> > > should feel comfortable asking for clarification, offering
> > > their own insights, and respectfully challenging ideas.
> > > Assuming that the teacher and student continue the
> > > interaction with one of these responses, which response
> > > will lead to a better interaction (do not let the order
> > > interactions affect your answer)? Output 1 or 2.

---

### Official Review · Reviewer_G1Y8 · 2024-07-15

**Soundness:** 3
**Presentation:** 2
**Contribution:** 2
**Rating:** 5
**Confidence:** 3

**Summary:**

The authors propose reinforcement-learning methods for multi-turn interactions using preference feedback in this paper. They present a mirror-descent-based policy optimization algorithm and prove its convergence to Nash equilibrium.

**Strengths:**

- The authors extend the RLHF paradigm to the multi-turn setting for an agent with multiple interactions with an external (stochastic) environment.

- In their feedback, they consider the entire multi-turn conversation and not just a single turn, which enables them to capture the long-term effect of individual actions and evaluate the conversation more accurately.

- The authors create and publicly release the Education Dialogue data.

**Weaknesses:**

- No human evaluation is performed, and the high-capacity LLM evaluation might not entirely reflect human preferences in a complicated conversation. Also, the prompt used for LLM evaluation only asks what conversation is better in general and doesn't compare different aspects of a good and valid conversation, such as consistency, compliance with the roles specified in the prompt, etc.

- Limited experiments only using small T-5 models and simple configurations.

**Questions:**

Refer to weaknesses.

**Limitations:**

The current evaluation is limited and can be extended to other models and tasks to show the effectiveness of the approach.

---

> ### Author Rebuttal · Authors · 2024-08-06
>
> We thank the reviewer for taking the time to evaluate our work. Here are the responses to your comments:
>
> Human evaluation: We refer the reviewer to the discussion in the main response on human evaluation and alignment. We briefly repeat the main message there: The main goal of our experimental section is to test our algorithms against other algorithms in how good they are able to align with preference data.
> Unlike single-turn benchmarks which are based on data with real human-preferences, our Education Dialogue domain is based on Gemini-Ultra curated preferences.
> As a result, evaluating this alignment using the same Gemini-Ultra model as done in our new presented experiments is true to our goal of understanding how to align to preference data. While human-evaluation is always interesting, here it is actually just a proxy to alignment with the data.
>
> Regarding the prompt for LLM evaluation: We agree that in a real dialogue system, it is important to collect diverse data from raters that account for many attributes of what it means to be a good conversation. However, the goal of this paper is not to create a real dialogue system, but to develop *better algorithms* to be used for alignment. We believe that grounded theory and our experimental results suggest that MTPO should be heavily considered when trying to create such a real dialogue system or any other multi-turn domain. It is important to point out that other algorithm-focused works (RLHF/DPO/IPO/NashMD) are also using a single-preference signal that tries to holistically capture the task (e.g., summarization). This is again due to the fact that these papers are not intended to create a SOTA summarization model, but rather to provide new algorithms to train one.
>
> Regarding model sizes: We added experiments with T5-XL (3B) that solidify the conclusions of our original experiments and we plan to add additional experiments with T5-XXL (11B) for the camera ready version.
> We refer the reviewer to the general response for a detailed discussion of our results.
>
> In light of the above, we kindly request the reviewer to reconsider their score based on both the significance of our theoretical contribution and the new experiments showing that our results extend to larger models.

---

> > ### Comment · Reviewer_G1Y8 · 2024-08-12
> >
> > Thank you for the thorough response. After reading the rebuttal, I decided to increase my score by +1.

---

### Author Rebuttal · Authors · 2024-08-06

We thank the reviewers for the time and effort put into the reviews. The following address points that appeared in multiple reviews.

# Paper Scope:
It appears to us that the reviewers have dedicated most of their attention to the experimental results, potentially overlooking the paper’s main contributions, i.e., introducing fundamental and grounded novel algorithmic concepts and a new multi-turn setting for LLM alignment. We emphasize that the purpose of our experiments is to validate our approach and theoretical findings in a practical setup, rather than to obtain SOTA.

Concretely, we kindly ask the reviewers to re-consider their judgment of the paper in light of the following main contributions:
1. We introduce the multi-turn setting for LLM alignment, which judges agents based on their overall communication capabilities instead of just their ability to provide an immediate useful response. This is crucial for enhancing LLM capabilities in scenarios, such as planning and complex tool-use. Moreover, we mathematically formalize this setting and define the novel objective of finding the policy that is best aligned with human preferences.
2. We identify a novel preference-based non-symmetric Q-function as the fundamental mathematical object that allows us to efficiently solve the multi-turn objective. Furthermore, we use this Q-function to come up with novel algorithms that are theoretically grounded and practical to implement. We also extend (and analyze) RLHF to the multi-turn setting.
3. Theoretically, we prove that our algorithms converge to the Nash equilibrium of the multi-turn setup (while clearly single-turn approaches fail to do so).
4. Empirically, we show that multi-turn approaches beat single-turn baselines.
5. We create a new preference-based multi-turn experimental environment. Since we are not aware of any such an environment, this important contribution gives the community a benchmark to further explore multi-turn alignment algorithms and improve LLMs multi-turn capabilities in the future.

# On (Human) evaluation and Alignment

The ultimate goal of learning from preferences is human alignment, and our multi-turn preference based algorithm is explicitly designed to improve the alignment of LLMs. However, collecting quality human data is a very long and expensive process. This is especially true when one needs to evaluate the difference between long dialogues. While recommended when training real-world systems, in academic settings, it has become standard practice to use a highly capable LLM (Gemini/ChatGPT) as a proxy for human alignment (see DPO, RLAIF, NashMD, IPO).

Additionally, unlike some of the above works that train on data collected from real human preference but evaluate performance with an LLM as a judge, in our work, the training preference data itself is generated by an LLM. Therefore, the true goal in our curated environment is to align the model with the preference of this highly capable LLM rather than a human rater.
Following the reviewers’ recommendations, we validated our results using the same Gemini Ultra model used to generate the data, reaching the same conclusions (see below) as with the prompted T5-xxl model used in the paper. This is indeed a more truthful measure for the model’s alignment . We wish to thank the reviewers for helping improve the paper, we will include these new and strong results in the final version.

**Evaluation with a Gemini-Ultra judge**
||SFT|ST RLHF Reward|ST RLHF Value|MT RLHF|MTPO|MTPO-tau|
|:------------------|:-----:|:--------------:|:-------------:|:-------:|:-----:|:--------:|
|SFT|-|0.206|0.286|0.164|0.125|0.086|
|ST RLHF Reward|0.794|-|0.479|0.452|0.447 | 0.277|
|ST RLHF Value|0.814|0.521|-|0.467|0.438|0.320|
|MT RLHF|0.836|0.548|0.533|-|0.419|0.288|
|MTPO|0.875|0.553|0.562|0.581|-|0.305|
|MTPO-tau|0.914|0.723|0.680|0.712|0.695|-|

# On the usage of the t5-large model:
We would like to point the reviewers’ attention to several important points regarding the use of T5-large models:
1. The multi-turn setting is more computationally demanding than single-turn, making the choice of smaller models especially reasonable in the context of an academic paper. Due to the repeated interactions, multi-turn dialogue is a much longer NLG task than common single-turn benchmarks like OpenAI TL;DR, Anthropic-HH, and XSUM, with an overall generation length of 20 times longer. Therefore, while we agree with the reviewers that comparing our algorithm on larger and more capable models is valuable, this was not feasible in this kind of academic research both from time and cost perspectives.
2. Our goal is to compare different algorithmic concepts to validate the benefits of our novel approach, and not to establish SOTA results. Thus, our comparison is valid and fair, as all baselines use the exact same model. While scaling up model sizes provides better performance, this is orthogonal to what our experiments aim to test.
3. Very well known concurrent work with similar experimental goals use the same T5-large models: The papers NashMD, IPO, online IPO, RLAIF, and more, use an even simpler experimental setup and have recently been published in top-tier conferences. This shows that T5 models are regarded as common practice for testing fundamental algorithmic concepts in academic contexts.
4. Per the reviewers request, we are currently running experiments with larger models. We note that these experiments take time to run but initial results on T5-XL (3B, see below) show similar gains for our multi-turn approach, and we will include the final results in the final version.

**Evaluation of T5-XL (3B) using a Gemini-Ultra judge**
||SFT|ST RLHF Reward|MT RLHF|MTPO-tau|
|-------------------|:-----:|:--------------:|:-------:|:--------:|
| SFT |-|0.295|0.101|0.041|
| ST RLHF Reward|0.705|-|0.180|0.069|
| MT RLHF|0.899|0.820|-|0.139|
| MTPO-tau| 0.959 | 0.931| 0.861|-|

* **MTPO-tau L vs. MT RLHF XL:** 0.525

---

### Author Response · Authors · 2024-08-12
**Progress of additional Gemma experiments**

Dear reviewers,

A brief update:
Together with the T5-XL experiments we already posted in the rebuttal and reaffirm our experimental results, we are working on adding a similar set of experiments with **Gemma models**.

We already (supervised) fine-tuned Gemma models for the agent and environment, used them to generate data for learning the reward and preferences model, and currently the reward and preferences training is underway. Once it's finished, we will train and evaluate MTPO and our multi-turn and single-turn baselines.

**We will add these results to the camera-ready version.**

Sincerely,
The authors

---

### Decision · Program_Chairs · 2024-09-25

**Decision:**

Accept (poster)

**Comment:**

This paper explores preference learning in multi-turn settings (where the reward is only provided after several turns). The authors propose an algorithm with theoretical converge guarantees, and show its effectiveness on a new multi-turn dialogue task.

Reviewers' initial impressions were mixed, with in particular concerns regarding somewhat unconvincing empirical results (limited to very small models, and on a potentially debatable experimental setup). However, the authors provided additional convincing results on "less small" models (and committed to include more later) as well as on an improved experimental setup, which I believe addressed those main concerns (and after the discussion period, all reviewers switched to a rating above the acceptance threshold).

I also want to show appreciation for the theoretical contributions of this work, which may be based off previous work but are still novel w.r.t. their application to the multi-turn setting, which I believe is indeed quite relevant to explore.

As a result, I am recommending acceptance. My request to the authors, though, is to include in the camera-ready the new experiments they promised to include, even if results end up not being as good as hoped.